# A Metric-Based, Meta-Analytic Appraisal of Environmental Enrichment Efficacy in Captive Primates

**DOI:** 10.3390/ani15060799

**Published:** 2025-03-11

**Authors:** Luke Mangaliso Duncan, Neville Pillay

**Affiliations:** 1Department of Psychology, University of Warwick, 6 University Rd, Coventry CV4 7EQ, UK; 2School of Animal, Plant and Environmental Sciences, University of the Witwatersrand, 1 Jan Smuts Avenue, Braamfontein, Johannesburg 2000, South Africa; neville.pillay@wits.ac.za

**Keywords:** primate, animal welfare, captivity, animal management, environmental enrichment

## Abstract

Non-human primates (hereafter ‘primates’) are maintained in a range of captive environments for reasons ranging from conservation to clinical research. An animal management tool that is widely used to ensure good welfare in these contexts is environmental enrichment (hereafter ‘enrichment’). Enrichment takes many forms, including adding new features to an enclosure and providing opportunities for social interaction or training activities, and is extensively regarded as beneficial for captive animals generally, including primates. However, enrichment is not always effective and can result in mixed or even detrimental outcomes. In order to identify those enrichment approaches that work best, and why others fail, we conducted a systematic review of the existing literature on primate enrichment. We developed and implemented a new metric for assessing the effects of enrichment on primates and used this to compare different enrichment approaches as reported in the literature. We found that different enrichment approaches generate varying degrees of benefit for captive primates, with training-based enrichment being the most beneficial. Primates housed alone seem to derive the most benefit from enrichment and the social system of the primate species in nature may also influence how beneficial enrichment is. We summarize the reporting of enrichment in the literature, identify existing knowledge gaps and highlight areas for future research.

## 1. Introduction

Studying animals in captivity provides many advantages relative to studying free-living populations [1] but is also burdened by various issues. Studying captive animal behaviour and physiology provides unique opportunities to examine processes and mechanisms that might not be possible under natural conditions or where a high degree of environmental control is necessary, such as in clinical research, but may not be entirely representative of a natural state [2]. From a scientific standpoint, there is a need to ensure the validity of results and their applicability in a broader context [3,4] as well as comparability to free-living populations [5]. Further to the issues of scientific validity, sustaining captive animal populations can be challenging in practical terms.

All species require certain conditions in which to survive and thrive in captivity. Non-human primates (hereafter ‘primates’) present a specific set of challenges for their captive maintenance [6]. Aspects of their biology cannot easily be replicated under captive conditions (e.g., broad diets in nature: [7], the fission–fusion social dynamics of free-ranging populations: [8]). Primates exhibit complex cognition and social relationships (discussed by [9]), traits which make them vulnerable to psychological and social stress under captive conditions [10], despite their high degree of behavioural flexibility [11]. Captivity also imposes a unique set of unnatural conditions on animals [12]. Consequently, they may experience frustration, anxiety, depression, helplessness and physical maladies. The stress of such conditions may lead to abnormal behaviour, physical stress, psychological distress, reduced reproduction and disease [13]. It is for these reasons that maintaining primates in captivity presents an ethical and management dilemma [14].

One of the tools available to address the problems of maintaining primates in captivity is environmental enrichment. It is generally accepted that the aim of environmental enrichment (also termed behavioural enrichment or behavioural engineering; hereafter ‘enrichment’) is context-specific. Thus, it follows that enrichment is broad and multifaceted; referring to a range of interventions in captive environments intended to improve animal welfare [15]. Enrichment typically provides behavioural opportunities which would otherwise be absent [16], encourages species-typical behaviour [4] and promotes positive affective states while limiting opportunities for negative affect [17]. Several authors have described diverse categorizations of enrichment approaches [18,19,20], highlighting the variety of interventions presented in the literature. However, these categorizations may fail to capture the breadth and nuance of enrichment, which ranges from introducing toys and other objects [21,22] through providing artificial grooming partners [23] to playing music to primates [24,25].

Enrichment was first applied as early as 1911 and was investigated in earnest through the latter half of the twentieth century [15]. The research into enrichment to date has suggested that it has a measurable impact on reproduction (reviewed by [26]), improves animal health [27] and immune function [28], induces positive cognitive bias [29] and improves behavioural coping [30]. Moreover, enrichment may even improve the outcomes of scientific studies not primarily aimed at evaluating enrichment [4]. There is a general understanding that enrichment is beneficial and hence of value [31] and that it fulfils needs which are unmet for captive animals [32].

While there are many examples of the beneficial effects of enrichment, not all enrichment is considered effective in reducing apparent stress or improving welfare as measured behaviourally [33,34] or physiologically [35]. Furthermore, many studies report mixed results when implementing enrichment. For example, Fuller et al. [36] reported that the overall activity budget of Wolf’s guenons (*Cercopithecus wolfi*) approached that of their free-living conspecifics but that enrichment had a limited impact on levels of aggression and socio-positive behaviour. In Western gorillas (*Gorilla gorilla*), auditory enrichment in the form of rainforest sounds, has produced beneficial [37], mixed [38] or no discernable effect [39] on groups housed at different institutions, suggesting that contextual variation may influence the outcome of enrichment interventions. Certainly, species differences also influence enrichment outcomes; for example, brown lemurs (*Eulemur fulvus*) exhibited reduced stereotypic behaviour whereas Mona monkeys (*Cercopithecus mona*) exhibited increased stereotypic behaviour in response to the same feeding enrichment devices [40]. Such findings suggest that the context of enrichment application influences enrichment outcomes and that current enrichment approaches cannot be used as a one-size-fits-all solution to the stresses of captivity.

Given that, while predominantly beneficial, enrichment does not always have the intended effect, it is important to consider what factors affect enrichment outcomes and how they do so. When considering enrichment practice generally, one major criticism is that it tends to be applied on a trial-and-error basis [19,41]. Additionally, as illustrated through auditory enrichment in Western gorillas, there may be a fallacious perception that if an enrichment protocol has worked in one context, it will work in another. In day-to-day practice, caregivers typically employ those enrichment tools they consider to be effective [32], rather than empirically testing enrichment protocols and the mechanistic dynamics of their application. This is unsurprising, given that organizations rarely possess the personnel and financial resources to apply rigorous investigations of enrichment efficacy [42]; in most institutions, enrichment is approached as an integral component of daily husbandry, rather than empirically tested for efficacy. However, there is a need for a predictive approach to environmental enrichment research (i.e., an approach which links interventions and outcomes in a consistent manner, to reliably predict how enrichment will affect an animal’s welfare before implementation) in order to increase the relevance and efficacy of enrichment practice [16]. While many researchers have already made great strides in improving the science of environmental enrichment, further improvement could only make it more structured, objective and reproducible and reduce redundancy, through unnecessary or wasteful utilization of resources that are often already limited. Some studies have attempted to apply predictive approaches to enrichment success (e.g., [43]) but far more research is needed. In order to shift enrichment research into a predictive domain, the first step is to identify those enrichment approaches that are effective and why others fail.

The majority of enrichment research has focused on rodent subjects [44] and, while some descriptive reviews of the topic are available [45,46], to the best of our knowledge no meta-analytic investigation of the efficacy of environmental enrichment approaches for captive primates exists. In part this is because to date, there is also no standardized welfare-centred metric of enrichment efficacy that can be used across enrichment types to assess the relative welfare impact of interventions. Our study aim was to appraise current enrichment practices and identify those that are most effective for primates generally. Adopting a meta-analytic approach, we identify relevant peer-reviewed literature available through online academic databases. We extract information about methodology, experimental design, species and environmental context from relevant publications. We then describe trends in the reported practice of environmental enrichment for captive primates and assess enrichment approaches using a comparative metric we devised. Using this novel metric, we explore the variation in efficacy associated with different types of enrichment and different enrichment contexts. Through this approach, we hope to provide a baseline for the development of predictive approaches to enrichment for captive primates and a metric tool for the comparative reporting and assessment of enrichment more broadly.

## 2. Materials and Methods

### 2.1. Literature Review and Data Consolidation

The review and meta-analysis adhered to the PRISMA 2020 guidelines for reporting of systematic reviews [47] and the approach taken is presented in Figure 1. Given the specialized scope of this meta-analysis, specific online databases were targeted in order to maximize the likelihood of finding relevant published information and streamline data collection. The databases included were identified based on the presence of articles that focused on or addressed aspects of animal welfare, animal behaviour, animal cognition, comparative psychology, primatology and veterinary science: BioONE, Directory of Open Access Journals (DOAJ), EBSCOhost Academic Search Complete, IngentaConnect, Oxford Academic Journals, ScienceDirect, Springerlink, Taylor & Francis Online and Wiley. The index of the Universities Federation for Animal Welfare (UFAW) journal ‘Animal Welfare’ was searched independently because it did not appear to be indexed in the archives listed above. In order to search the literature for applicable studies, the Boolean phrase “primate AND environmental enrichment OR behavioural enrichment OR behavioral enrichment OR behavioural engineering OR behavioral engineering” (or a variation thereof, depending on what was permissible as search criteria for each database) was used and only peer-reviewed articles in English published up to 31 December 2023 were considered; no lower threshold date was implemented in order to maximize the number of hits for the search such that the search would identify any peer-reviewed articles available up to 31 December 2023.

We assumed that all published studies were independent and that no misrepresentation or reuse of data (i.e., the same data presented in two independent publications) was present. A control or baseline condition was a prerequisite for inclusion in this meta-analysis because studies lacking a control or baseline condition had no reference point from which to assess enrichment efficacy. However, studies which contrasted more than one housing environment or social condition without serving as a control, the least complex and/or stimulating for the animals (based on the descriptions provided) and/or the smaller environment (in respect to housing conditions) was assumed as the control condition. Studies that explicitly manipulated spatial and social density in captivity in order to investigate the effects of crowding were omitted; primates respond to spatial and social crowding through various behavioural and social tactics and the interpretation thereof in the context of welfare is complex and equivocal [48]. Statistical analyses of quantified welfare measures (e.g., behaviour, hormone levels) were also a prerequisite for inclusion in the meta-analysis such that studies that quantified welfare-related parameters but did not apply formal statistical testing were not considered. However, for studies that applied formal statistical testing and reported measures of assessment that are difficult to quantify (e.g., qualitative behavioural change, decreased severity of self-wounding), these measures were assessed (see the efficacy index below).

Each article was assessed against the enrichment protocol(s) described therein. Enrichment protocols were considered distinct if they varied in any of the classifications identified according to the following:Communication type (case report, research article or other);Institutional context (laboratory, rehabilitation centre, sanctuary, zoo or other);Primate species;Social conditions (group housing, pair housing, solitary, mixed housing);Housing type (indoor only, indoor–outdoor, outdoor only, standard housing (standard housing was considered as indoor-only cage rack systems, which typically employ enclosing steel mesh barriers, excretia trays below a steel mesh floor, a feeder hopper and a water supply) or other);Housing state (enriched, barren);Access to enrichment (0–3 h, 3–24 h, 1–3 days, >3 days);Enrichment type (see Table 1).

Thus, if an article contrasted the effects of object-based enrichment and social enrichment, all things being equal, two protocols were considered for that article: an object-based enrichment protocol and a social enrichment protocol.

All classifications represented the state of the subjects and study environment prior to implementation of the experimental condition (i.e., the baseline phase/control condition). For all classifications except communication type, another classification state was included (‘Undisclosed’) to account for those studies that did not provide the full complement of information. Reported sample sizes (i.e., number of subjects involved per protocol) were recorded per species. Case reports were defined as articles that detailed effects of enrichment primarily in descriptive terms, typically involving a single subject and a clinical intervention or necropsy or involved little to no statistical analyses; such articles were excluded from the meta-analysis but are summarized in Appendix A. With regard to the housing state, an environment was classified as barren if it provided the minimum furnishing for biological functioning and safety (i.e., food, water and enclosing physical barriers); any additional furnishings would result in classification as ‘enriched’. Although these are conservative dichotomous criteria, they are akin to those used in the rodent literature (sensu standard housing: [49]).

Access to enrichment was assessed according to when the enrichment was directly available to the animals at any given time (i.e., from incorporation/start to removal/cessation/exhaustion), such that, for example, an experimental treatment involving 2 h of access to a feeding enrichment device daily for a three-week period would be classified as 0–3 h, rather than >3 days. Access was not considered in relation to the duration of the experimental phase of the study, the cumulative duration of access or frequency of presentation for several reasons. Firstly, while enrichment can have lasting effects, the enrichment is most salient to the animals during the immediate period that it is available to engage with. Secondly, many studies did not provide sufficient detail in order to determine the cumulative duration of enrichment access over the study period. Further to this point, utilizing the duration of the study period as a measure of access may be misleading in instances where enrichment is rapidly depleted or short-lived, such as foraging devices or training sessions. Thirdly, the frequency of presentation of enrichment was not considered because presentation frequency may be misleading when considering efficacy and is better suited to experimental, rather than meta-analytic, investigation. For example, a single social intervention, such as transitioning to housing individuals in groups, may provide marked benefit for primates while frequent social changes may be detrimental to primates, due to associated hierarchical instability [50]; by contrast, enrichment focused on multiple feeding opportunities is more beneficial than a single feeding enrichment event [51]. In both examples, the frequencies might be identical but the effects would be very different, providing little clear insight into the influence of frequency on enrichment efficacy. If information on the duration of access was not available or was ambiguous, it was assumed that the enrichment was available to the animals for the full duration of the respective phase of the study.

The methods of assessing welfare, and their associated measures, varied across the identified publications and protocols. Thus, for each protocol and each reported measure, the study outcomes were scored using a three-point scale where a negative outcome (i.e., an outcome indicative of reduced welfare as per the criteria in Table 2) was given a score of −1 and a positive outcome (i.e., an outcome indicative of improved welfare as per the criteria in Table 2) was given a score of 1. Any outcomes which presented no notable change or equivocal outcomes were given a score of 0. For studies that examined the effect of removing enrichment, the results were interpreted conversely (e.g., removing enrichment increased inactivity; thus, having enrichment reduced inactivity). To determine the effect of the enrichment, a score was assigned based on the conversely interpreted effect. The score was not scaled according to the reported extent of change in the outcome criterion but merely upon whether statistically significant changes occurred and the nature of the change (i.e., statistically significant positive change, no change or statistically significant negative change). If the outcome of a metric used in a given study was not reported in the results of the study or was reported ambiguously, it was assumed that the outcome was not statistically significant and a score of 0 was assigned.

Each protocol outcome was considered in relation to group-level outcomes of the enrichment protocol. Therefore, studies that reported individual-level outcomes, a mean score based on the outcomes per individual was recorded for that protocol. For example, if a study reported on the effect of enrichment on 5 subjects, the index score for each subject was calculated and the mean of these scores was then used. This was performed to facilitate direct comparison between studies (it is easier to generalize individual-level outcomes to a group-level than vice versa) and to examine the data in a way that would be most applicable to the broader populations of captive primates, given the idiosyncrasies of individual’s responses to enrichment. Similarly, for protocols that examined several variations of a specific enrichment design (e.g., a protocol that applied different food rewards to the same puzzle-feeder to contrast the effects of different reward types), a mean score based on the individual outcomes associated with each variation of the specified enrichment design was assigned to the protocol. Each protocol was assessed only for those criteria that were relevant to the protocol in question; for example, if a protocol reported on behaviour only, scores were not assigned for neurocognitive, physiological and reproductive criteria.

The following equation was then used to calculate an efficacy index for each enrichment protocol:$$E=\frac{\sum_{i=1}^{n}w_{i}s_{i}}{n}$$
where *n* represents the number of outcome measures reported in the protocol (e.g., for 5 behavioural and 2 hormonal measures: *n* = 7), *w* represents a weighting factor (in this study all outcomes were equally weighted so as not to introduce bias into the sample used here and thus the weighting factor was 1; the weighting factor is considered further in the Section 4 below), *s* represents the given outcome measure score (−1, 0 or 1) and *E* represents the efficacy index score. The resulting efficacy index score ranges from −1 to 1 whereby, for a given enrichment protocol and set of outcome criteria, a score of −1 represents diminished welfare and a score of 1 represents enhanced welfare for all measures assessed. Thus, for example, if a study reported reduced abnormal behaviour, increased social interaction, no change in activity but increased aggression, each behaviour would be scored as abnormal = 1, social interaction = 1, activity = 0, aggression = −1, and the sum of the criteria scores (1 + 1 + 0 + (−1) = 1) divided by the number of reported outcome criteria (in this instance, 4 behaviours) would result in an efficacy index of 0.25 for that specific enrichment protocol.

Consensus regarding what does and does not constitute ‘desirable’ behaviour in captive animals is lacking and interpretation of the welfare implications of behaviours is problematic, particularly for ‘undesirable’ natural behaviours (e.g., aggression, predation and territoriality). Furthermore, interpreting behaviours collectively is difficult, especially when some important behaviours are affected by the application of enrichment while others are not. For these reasons, behaviours were assessed independently of one another according to their interpretation as desirable by the authors of the protocol under assessment and in relation to the criteria outlined in Table 2. Similarly, changes in other somewhat subjective estimates of welfare, such as behavioural diversity, body condition or reproductive abnormalities, were scored according to the interpretation of the authors regarding their welfare significance. In the case of behavioural diversity, behaviours identified as being of welfare significance by authors were treated as a single outcome measure such that if multiple new behaviours arose in response to an enrichment protocol, a single score was assigned to account for increased behavioural diversity.

The natural history of a species may predict their responses to captive life. Clubb and Mason [52] illustrated this principle in carnivores by showing that ranging patterns in nature explained the propensity for the onset of stereotypy under captive conditions. In addition, Queiroz and Young [53] demonstrated that the natural ecology of a species influences how they respond to the presence of zoo visitors. Thus, for each primate species, we gathered data on the natural habitat and ranging patterns of free-ranging conspecifics using the International Union for the Conservation of Nature (IUCN) Red List website (http://www.iucnredlist.org/; date accessed: 30 April 2024), Animal Diversity Web (https://animaldiversity.org/; date accessed: 30 April 2024) and/or Primate Info Net (https://primate.wisc.edu/primate-info-net/pin-factsheets/; date accessed: 30 April 2024). Habitat data were classified into broadly defined global habitat types, namely subtropical/tropical forest, temperate forest, savanna, grassland, rocky, shrubland, wetland and anthropogenic/disturbed habitats. For each species, the number of different habitat types occupied by free-ranging populations was recorded as a measure of the diversity of habitats occupied by the species in nature. Data of maximum and minimum home range (in km^2^) were also recorded from the above-mentioned databases. Data for the home ranges of *Gorilla gorilla* [54], *Macaca arctoides* [55], *Macaca silenus* [56], *Nycticebus bengalensis* [57], *Papio anubis* [58], *Papio hamadryas* [59], *Papio papio* [60] and *Sapajus xanthosternos* [61] were supplemented using the reported publications because data were not available or incomplete elsewhere. Home range data for the genus *Galago* were used for *Galago senegalensis* because only genus-level data were available. Where only average home range data were available, these were used as the respective maxima and minima. Given the fragmentary data on the ecology of the species [62], no data appear to exist for home range sizes of *Mandrillus leucophaeus* [63]. For *Nycticebus bengalensis*, no clear data on group sizes are available (sources state merely that they may be solitary or live in small family groups) and thus maximum group size was conservatively estimated to be 2, representing a female and her single offspring. The IUCN Red List website was also used to confirm and update species names and confirm species’ conservation status because some studies used in this meta-analysis did not use contemporary classifications. Each species was also classified by family and according to broad phylogenetic grouping (Prosimian, South American monkeys, Afro-Asian monkeys and Apes).

### 2.2. Data Analysis

Of the identified articles, none of the case reports or ‘Other’ publications provided sufficient information for inclusion in the analysis. Thus, the final dataset comprised only environmental enrichment protocols from published peer-reviewed research articles. Studies which pooled data from different species, but which otherwise met the criteria for inclusion in the meta-analysis, were excluded because these rendered species-level differences in response to enrichment impossible to distinguish. Only three protocols involved primates in a rehabilitation centre setting and only two were carried out in a primate sanctuary setting. These data points were excluded from statistical analyses because of the impact their inclusion would have on statistical power for analyses and the likelihood of emergent false negatives; descriptive information for these studies is provided however (Appendix A). A further two protocols, which otherwise met the necessary criteria for inclusion, did not clearly disclose sample sizes and were thus also omitted from the dataset used for analyses but are included in the descriptive summaries provided in the results.

A principal component analysis was run to identify a potential correlation between predictor variables in the full dataset, using institutional context, social conditions, enclosure type, housing state, enrichment type, enrichment access duration, sample size, broad phylogenetic group, family, habitat types and maximum and minimum home range data and group size in nature. We found that broad phylogenetic grouping and family were associated (i.e., autocorrelated) and thus only family was considered in further analyses. No other correlations were identified between variables.

In order to investigate the predictors of environmental enrichment efficacy, the data were initially analyzed using two generalized linear mixed models using the lmer function in the lme4 package (ver. 1.1-35.5) [64] in R ver. 3.5.1 [65]. The first model encompassed the species biology and natural ecology of the primate species in relation to the efficacy of enrichment (hereafter the bio-ecology model) and the second model encompassed aspects of the captive environmental context in relation to enrichment efficacy (hereafter the captivity model). Using AIC comparisons, the exclusion of random factors (i.e., subject species and relative sample size per protocol) in the models was found to improve the model fit (Bio-ecology model AIC: 286 VS 168; Captivity model AIC: 269 VS 172). Thus, the final models were run using the glm function in the stats package (ver. 4.2.2). Factor significance was assessed through analysis of deviance type III Wald χ^2^ testing using the Anova function in the car package (ver. 3.1-3) [66] in R. Given the number of factors under investigation, only first-order effects were considered. All analyses were two-tailed and results were considered as significant for *p* ≤ 0.05. Significant differences between within-factor levels were identified using β-estimates and confidence intervals and are reported as absolute value β-estimates, standard error and *p*-values. For significant continuous predictors, Spearman’s rank correlation test was run using the cor.test function in the stats package (ver. 4.2.2) to assess the relationship between predictors and enrichment efficacy scores.

For all models, the efficacy index scores were first transformed by adding 1 to each score in order to ensure all values were greater than zero, while preserving the score’s relative meaning, and were used as the response variable. For the bio-ecology model, the categorical predictors included family and enrichment type and the continuous predictors included number of habitat types occupied by a species in nature and respective maximum and minimum home range and group sizes. Because no data exist on the home ranges of *Mandrillus leucophaeus* [63], the two data points for the species were omitted from the bio-ecology model. For the captivity model, the categorical predictors included institutional context (i.e., laboratory, zoo, other), social context (i.e., group housing, pair housing, solitary, mixed housing), enclosure type (i.e., indoor only, indoor-outdoor, outdoor only, standard housing or other), housing state (i.e., enriched, barren), enrichment access duration (i.e., 0–3 h, 3–24 h, 1–3 days, >3 days) and enrichment type.

## 3. Results

### 3.1. Summary of Findings in the Literature

A total of 172 articles, distributed across 21 peer-reviewed journals, were identified as relevant, met the criteria for analysis and were included in the sample used for the meta-analysis (see Appendix A). These articles yielded 248 independent enrichment protocols applied to 55 primate species (Table 3, Appendix A). The temporal distribution of enrichment study protocols used in this meta-analysis is shown in Figure A1. Despite an increasing trend in publication of primate enrichment research, the global COVID-19 pandemic in 2019 was associated with a marked drop in published primate enrichment research (Figure A1).

Zoo studies accounted for 45.56% of protocols, laboratory settings accounted for 44.35% of protocols included in our sample and the remaining protocols were conducted in sanctuaries (0.81%), rehabilitation centres (1.61%) or undisclosed institutional contexts (7.66%). Studies identified in the context of primate rehabilitation centres and sanctuaries that were not included in statistical analyses are detailed as ‘Other reports’ in Appendix A along with the case studies identified from the literature.

There was considerable variation in baseline conditions. When considering baseline social context, 52% of protocols involved group-housed subjects, 9% involved pair-housed subjects, 26% involved solitary subjects and 10% involved subjects across varying social conditions while 3% did not disclose the social conditions at the start of the study. Of the protocols under consideration, 17% provided indoor-only housing, 35% provided indoor-outdoor housing, 9% provided outdoor-only housing, 27% provided standardized laboratory housing, 2% provided mixed housing conditions and 10% did not disclose the housing type. Regarding housing enrichment state, 70% of protocols provided an enriched environment, 10% provided an impoverished environment, 1% drew from subjects in mixed impoverished and enriched conditions and 19% did not disclose the level of pre-existing enrichment in the baseline condition.

Methodologies varied across protocols with many employing multiple means of assessing welfare, including behavioural, physiological, cognitive, neuroanatomical, clinical and parity records and developmental assessments. Protocols that used only a single methodology primarily considered the impact of enrichment in terms of behavioural assessment (214 protocols), followed by physiological assessment (10 protocols), direct or record-based clinical assessment (3 protocols), assessment of developmental milestones and/or psychological development (3 protocols), cognitive assessment (1 protocol) and another method (food consumption as an estimate of energetic expenditure resulting from enrichment use by animals: 1 protocol). 16 protocols used multiple methods of assessment concurrently (up to four different methods in one protocol). Regardless of the number of assessment methods used, behavioural assessments were conducted in 229 protocols, physiological assessments in 23 protocols, clinical assessments in 6 protocols, developmental assessments in 4 protocols and cognitive assessments in 3 protocols.

For the enrichment access duration, 18.95% of protocols considered enrichment within a 0–3 h timeframe, 12.50% a timeframe of 3–24 h, 0.81% a timeframe of 1–3 days, 56.85% timeframes of more than 3 days and 10.89% did not disclose a clear timeframe. Feeding-based enrichment was the most prevalent enrichment approach in the literature while interaction-based enrichment was applied the least. Training-based enrichment generated the highest enrichment efficacy index scores and interaction-based enrichment generated the lowest efficacy index scores (Figure 2). A summary of efficacy scores for each enrichment type and species is provided in Appendix A.

The majority of research was conducted on species of least conservation concern, followed by critically endangered species (Table 3). Appendix A provides a taxonomic summary of the number of protocols and sample sizes reported in the literature and the associated efficacy scores. Rhesus macaques (*Macaca mulatta*) and chimpanzees (*Pan troglodytes*) comprised the focal species for the bulk of published studies. For all primate species, the average enrichment efficacy score (±SE) was 0.22 ± 0.02. The Lorisidae had the highest efficacy of enrichment but was the least-studied group, and the Callitrichidae had the lowest efficacy score. When considering efficacy at a species level, the highest enrichment efficacy scores calculated were scores of 1 for black-tufted ear marmosets (*Callithrix penicillata*), black capuchin monkeys (*Sapajus nigritus*) and Northern white-cheeked gibbons (*Nomascus leucogenys*), whereas the lowest efficacy score was generated for Abyssinian colobus monkeys (*Colobus guereza*: −0.22). However, in all cases relating to the highest and lowest enrichment efficacy scores listed above, only a single enrichment protocol was published for each species, suggesting that these scores may not be an accurate reflection of the enrichment efficacy for the respective species.

### 3.2. Errant Trends in Reporting

Numerous studies did not report sufficient information for inclusion in this meta-analysis. Several of these lacked a baseline or reference treatment against which results could be compared (e.g., [67]), many of which contrasted variations of an enrichment protocol (e.g., [68]) or housing condition (e.g., [69]) or investigated the preferences of subjects for specific types of enrichment [70]. Many studies reported on levels of engagement with enrichment by animals rather than its impact (e.g., [67]) or, surprisingly, did not provide a value conclusion (i.e., did the enrichment improve animal welfare or achieve what was intended?) in relation to the effect of the enrichment protocol(s) under consideration, even if the explicit aim was to assess enrichment efficacy (e.g., [71]). In some studies, meaningful outcomes could not be extracted because subjects of different species were analyzed as a single group (e.g., [72,73,74,75]), presumably to overcome sample size limitations, or the species and numbers involved were reported ambiguously (e.g., [76]). Many studies did not provide complete information regarding the enrichment access duration (e.g., [77]) or housing conditions, with approximately a third of studies used for this meta-analysis failing to disclose housing conditions completely (i.e., social conditions, housing type or housing state).

Data processing and statistical analyses were rarely completely described, leaving readers to intuit what data comprised the final dataset and how it was analyzed (e.g., [78]). Some conducted no statistical analyses (e.g., [79,80,81]) and some studies described statistical model fit through comparison of Akaike Information Criterion (AIC) values without describing the statistical effects of enrichment interventions (while statistically valid, such an approach makes interpretation of the findings challenging and ambiguous, limiting applicability to the field more broadly; e.g., [82]). Reporting of statistical findings is inconsistent across the literature with some studies reporting precise *p*-values (e.g., *p* = 0.022) and others reporting statistical significance in a categorical fashion (e.g., *p*-values greater than 0.05 are reported as “NS” and *p*-values less than 0.05 are reported as falling into either “*p* < 0.05”, “*p* < 0.01” or “*p* < 0.005”: [83,84]); such practice fails to conform to widely held standards of statistical reporting [85] and is particularly problematic for statistically non-significant results because categorical reporting may mask biologically meaningful effects or trends which are not strictly statistically significant. Other studies reported only *p*-values, with no test statistics or other statistical metrics (e.g., [86]) and/or omitted results for non-significant effects (e.g., [87]). A mismatch between reported metrics (usually behaviours) in the methodology and those reported in the results was also common (e.g., [77,84]) and complicated interpretation of the study outcomes; many studies described detailed ethograms in the methodology but provided results for different categories of behaviour without an explanation of how these categories were translated from the ethograms described. Almost no studies reported diminished welfare in subjects as a result of enrichment interventions or documented specific interventions that failed to generate a desirable effect on subjects (but see Appendix A).

### 3.3. Statistical Models

When considering the data presented in the literature, only minimum group size was identified as a significant predictor of enrichment efficacy scores in the bio-ecology model (χ^2^_1_ = 4.28, *p* = 0.038) but the relationship between minimum group size and enrichment efficacy scores was weakly positive (ρ = 0.07, *p* = 0.278), suggesting that primates with larger minimum group sizes in nature may benefit more from enrichment than species with smaller minimum group sizes in nature. None of the remaining predictors of enrichment efficacy scores were found to be significant in the bio-ecology model (enrichment type: χ^2^_12_ = 15.72, *p* = 0.205; family: χ^2^_7_ = 11.12, *p* = 0.133; number of habitats occupied in nature: χ^2^_1_ = 1.06, *p* = 0.304; maximum home range: χ^2^_1_ = 0.00, *p* = 0.950; minimum home range: χ^2^_1_ = 0.22, *p* = 0.642; maximum group size in nature: χ^2^_1_ = 0.42, *p* = 0.515).

In the captivity model, enrichment type was a significant predictor of enrichment efficacy scores (Figure 2: χ^2^_12_ = 21.40, *p* = 0.045) as was social context (χ^2^_4_ = 10.61, *p* = 0.031). Institutional context (χ^2^_2_ = 0.52, *p* = 0.768), enclosure type (χ^2^_6_ = 6.49, *p* = 0.370), housing state at the start of the study (enriched or barren: χ^2^_3_ = 0.26, *p* = 0.968) and duration of application of enrichment (χ^2^_4_ = 4.41, *p* = 0.353) were not significant predictors of enrichment efficacy.

Training-based enrichment generated the highest efficacy index scores whereas enrichment involving heterospecific interactions generated the lowest efficacy index scores (Figure 2). Significant differences between the efficacy index scores of specific enrichment types were found between auditory and cognitive (mean index scores: 0.02 vs. 0.31), auditory and feeding (0.02 vs. 0.30), auditory and training (0.02 vs. 0.35), cognitive and interaction (0.31 vs. −0.11), cognitive and object (0.31 vs. 0.09), cognitive and social (0.31 vs. 0.21), enclosure modification and interaction (0.26 vs. −0.11), feeding and interaction (0.30 vs. −0.11), feeding and object (0.30 vs. 0.09), interaction and training (−0.11 vs. 0.35), object and training (0.09 vs. 0.35) and social and training (0.21 vs. 0.35) enrichment interventions (see Appendix A for β-estimates and *p*-values). All other contrasts yielded no significant differences between enrichment types (Appendix A).

As regards the social context of enrichment application, the highest enrichment efficacy scores were obtained in those studies where the social context was undisclosed and the lowest scores were obtained in studies focusing on group-housed subjects (Figure 3). Significant differences in the enrichment efficacy scores were found between group and solitary housing contexts (mean index scores: 0.18 vs. 0.30; β = 0.18 ± 0.08, *p* = 0.024) and between the group and undisclosed housing contexts (0.18 vs. 0.51; β = 0.36 ± 0.15, *p* = 0.015). No significant differences emerged between group and mixed housing (0.18 vs. 0.22; β = 0.10 ± 0.09, *p* = 0.227), group and pair housing (0.18 vs. 0.24; β = 0.05 ± 0.08, *p* = 0.549), mixed and pair housing (0.22 vs. 0.24; β = 0.05 ± 0.10, *p* = 0.600), mixed and solitary housing (0.22 vs. 0.30; β = 0.08 ± 0.10, *p* = 0.436), mixed and undisclosed housing (0.22 vs. 0.51; β = 0.26 ± 0.16, *p* = 0.117), pair and solitary housing (0.24 vs. 0.30; β = 0.13 ± 0.10, *p* = 0.169), pair and undisclosed housing contexts (0.24 vs. 0.51; β = 0.31 ± 0.16, *p* = 0.055) or solitary and undisclosed housing contexts (0.30 vs. 0.51; β = 0.18 ± 0.16, *p* = 0.271).

## 4. Discussion

The aim of this meta-analysis was to review the existing literature to generate a comprehensive appraisal of the trends in the reporting of the effects of environmental enrichment (hereafter ‘enrichment’) for captive primates. Our intention was to identify those enrichment designs that are most effective for primates generally and to explore the variation in efficacy associated with different types of enrichment and different enrichment contexts. The following discussion of our findings focuses on three areas: the context of enrichment application and reporting, the efficacy of enrichment practice and our efficacy index as an enrichment efficacy assessment tool.

### 4.1. Contexts of Application and Reporting of Enrichment

Several noteworthy trends emerge in relation to primate enrichment studies. Firstly, primate enrichment has received increasing attention since the earliest publications in 1981, a pattern which was apparently disrupted only by the COVID-19 pandemic (Figure A1). Several processes may be driving this trend. It may be the result of increasing awareness within scientific circles of the potential impact of the captive environment on the physiology, psychology and behaviour of primates and subsequent impact on scientific outcomes [88]. Similarly, the need to meet legislative and regulatory requirements for minimum standards of housing and husbandry (e.g., [89]) may be a potential contributing factor. External pressure stemming from the awareness and interest of broader society about animal welfare [90,91], the ethics of maintaining captive primate populations and/or the charismatic appeal of primates relative to other captive species [92] may also be contributing factors. A total of 110 of the 172 (64%) research articles considered in this meta-analysis were published in journals that explicitly focus on animal welfare or the application of animal welfare science (typically in a zoo context; Appendix A). It therefore seems plausible that primate enrichment is considered more in relation to ethical concerns and husbandry than its impact on science practice or scientific validity. While the scientific implications of enrichment interventions are likely to be more consequential in a research context than a husbandry context, a strong argument can be made for the need to ensure scientific validity in captive animal research generally [4]. While the increase in attention to primate enrichment is encouraging in welfare terms, there is an evident need for a broader interpretive perspective, specifically considering the scientific impact of primate enrichment.

Secondly, the phylogenetic distribution of primate enrichment studies indicates a distinct bias for some species. Two species in particular, rhesus macaques and chimpanzees, are the focus of most enrichment research. Rhesus macaques, which accounted for 18% of the protocols considered in this meta-analysis, are in high demand in biomedical research and constitute a large proportion of laboratory-housed primates worldwide (18% of reported studies: [93], 65% of laboratory primates in the United States of America: [94]). By contrast, laboratory chimpanzee populations have been steadily declining worldwide following legal restrictions on their breeding and utilization in laboratory-based research since at least 2008 [95]. Yet, they remain one of the most popular species exhibited in zoos [96] and subjects of zoo-based research [97]. Thus, the prevalence of rhesus macaques and chimpanzees in the enrichment literature may reflect their high occurrence in the laboratory and zoo contexts, respectively. Furthermore, these species are likely to be present across a wider range of captive contexts (i.e., they may be present in laboratories, zoos, sanctuaries or rehabilitation centres) than other primate taxa, such as gibbons or prosimians, which may be less likely to be found in laboratories, for example. Certainly, other species (e.g., pottos: *Perodicticus* spp.) are inherently rarer in captivity, possibly due to their nocturnal habit and venom, and may thus be less likely to constitute the focus of enrichment research. These factors likely contribute to this apparent phylogenetic bias in primate enrichment research. However, it is difficult to test the veracity of this assumption given that no complete record or estimates of worldwide captive primate population sizes or compositions are freely available.

Some species may be viewed as less charismatically appealing, of lesser conservation significance (thereby seemingly warranting less attention) or are less likely to generate funding for primate enrichment research. Fundraising-centred investigations suggest that both charismatic appeal and similarity to humans significantly impact on the perceived value or degree of interest in a species [98] and, to this end, species or families that are arguably more human-like (e.g., great apes) attract more interest [21]. This results in an overstated prevalence of these groups in the literature, as noted in a previous meta-analysis on enrichment in zoos [16]. Furthermore, the general public exhibits distinct species-viewing preferences [96,99] and influence how enrichment is practiced in zoos [100]. We found 60 published protocols that focused on four species within the Hominidae (an average of 15 protocols per species) whereas 107 published protocols focused on 22 species within the Cercopithecidae (an average of 4.9 publications per species). This indicates a proportional bias towards species within Hominidae ([16], also previously identified as a particularly popular group in enrichment research: [101]). Assuming that the distribution of species in the literature mirrors their prevalence in captivity, this potentially implicates charismatic appeal and ‘human-ness’ in determining whether a species is the focus of enrichment research.

It is intriguing that phylogenetic family did not emerge as a significant predictor in the statistical analysis. It may be that the natural ecology and behaviour of a given species has a stronger effect on welfare in captivity and how they respond to changes in their environment than might be accounted for by evolutionary relationships. Certainly, relatively closely related species or genera may have very distinct ecologies (e.g., geladas vs. baboons) and these species differences can significantly impact how animals respond to captive conditions [102]. Moreover, natural ranging behaviour has been shown to predict the expression of stereotypy in captive carnivores [52] and primates [103], independently of phylogenetic relationships. It is also important to acknowledge that in our analysis, some primate families were under-represented and thus the role of sample size in affecting the outcome cannot be ruled out.

As previously mentioned, studies of enrichment in the rehabilitation (two studies) and sanctuary (one study) institutional contexts were conspicuously scarce in the literature used for this study. Sakuraba et al. [104] illustrate the utility of enrichment in physical rehabilitation (discussed below) but enrichment has potential as a valuable tool for conservation and rehabilitation in other contexts as well. For example, food acquisition skills are imperative for primates to be released into nature [105] and enrichment protocols that require naturalistic food-processing techniques may facilitate important skills acquisition for later feeding challenges in a non-captive setting (e.g., providing unprocessed whole fruit for lemurs: [75], artificial termite fishing mounds in chimpanzees: [106], nut-cracking tasks in tufted capuchins, *Sapajus apella*: [107]). The dearth of literature on enrichment in sanctuaries is of particular concern considering that sanctuaries implicitly house animals for the remainder of their lifetime to ensure their physical and psychological well-being (e.g., Chimpanzee Sanctuary Uto: [108]). Many of these animals may exhibit behavioural, social, physical and psychological dysfunction [69] which might be ameliorated through enrichment. That is not to say that enrichment is absent in sanctuaries but rather that it is not reported in the scientific peer-reviewed literature. In accordance with these findings, a recent review of research conducted in primate sanctuaries made no mention of enrichment in this context [109].

There was considerable variation in the baseline conditions reported in the literature. The influence of baseline state, sometimes referred to as baseline severity, is well documented in the clinical research context [110]. It may be that animals which typically experience severe chronic stress respond better to interventions than those who experience milder or intermittent stress or, alternatively, may prove somewhat resistant to interventions due to the severity of the experienced stress. In laboratory mice, for example, cooler housing conditions generate chronic stress, which then influences experimental outcomes [111]. Evidence from this study suggests that the social conditions of captive primates may influence the outcome of enrichment interventions (see Section 4.2) and thus it is likely that variation in baseline severity may account for variation in enrichment effects. It is however difficult to envision an approach to enrichment in captive primates which might standardize baseline conditions, especially given the range of contexts in which they are maintained (laboratories, zoos, sanctuaries, rehabilitation centres, etc.) but future studies and enrichment interventions should consider the potential influence of baseline severity on enrichment efficacy.

Fraser [112] suggests that an integrated multidisciplinary approach is critical to generating a holistic understanding of the welfare state of a given animal, which will also apply to enrichment research. Enrichment can be assessed in a variety of ways [33]. Yet, there is currently a lack of diversity in the assessment methodologies employed in enrichment studies, with behavioural assessments predominating zoo-based primate research [101]. The findings of our meta-analysis echo this view; behavioural observation and recording was the primary tool used in the majority of studies to assess the impact of enrichment on subjects (92% of all protocols considered used a behavioural metric) and only 6% of the protocols utilized multiple means of assessing efficacy. Furthermore, the duration of application and assessment of enrichment in the literature vary considerably but tend to consider either very short or longer-term exposures to enrichment despite evidence that enrichment has a variable effect, depending on the duration of its application [46]. The approach to assessing welfare tends to depend on the adopted welfare philosophy (i.e., whether the focus is on health and physiological functioning, affective states or natural behaviour, mirroring the long-standing philosophical debate about what constitutes a ‘good’ life in humans: [112]). However, it is critical that alternative methods to quantify the effect of enrichment be used wherever possible. The aim of enrichment varies but generally seeks to encourage contextually adaptive behaviour [19], promote species-typical behaviour, promote animals’ control of their immediate environment and promote homeostasis [4]. Such a multifaceted and complex state of biological functioning (reviewed by [113]) is unlikely to be quantified using behavioural measures alone. In short, multiple, correlated lines of evidence considering both short- and long-term effects will likely provide a far more holistic understanding of the impact of enrichment than studies of limited contextual and temporal scope.

The possible exception to the above is case reports (not considered in our analysis) that provide valuable information about environmental enrichment effects that might evade detection or mention in a strictly experimental context. Of particular importance are secondary effects of enrichment (i.e., emergent unintentional or indirect effects of enrichment), which may be detrimental, neutral or beneficial. Three case reports pertaining to serious detrimental secondary effects of enrichment were identified during this meta-analysis. These were three instances of intestinal perforation or blockage as a result of ingestion of enrichment materials, unfortunately leading to euthanasia of the affected individuals [114,115,116]. Gastrointestinal blockages in captive primates are well documented (e.g., [117]) because primates tend toward oral exploration and play [115]. Injury risk is sometimes cited as a reason for not providing enrichment in captivity (*Pers. obs.* L. Duncan) but it is noteworthy that the detrimental effects of enrichment are typically restricted to a single individual [116]. In two of the three case studies, the authors concluded that the beneficial enrichment effects superseded the potential harm to individuals [115,116].

Secondary effects of enrichment may also be desirable or beneficial. Some case reports provide detailed descriptions of the emergence of novel behaviours upon presentation of enrichment [118]. Jones and Pillay [119] describe the stimulation of appetitive food searching behaviour in zoo-housed hamadryas baboons (*Papio hamadryas*) as a secondary effect of observing the monopolization of a feeding device by a dominant individual. Therefore, enrichment can have subtle group-level effects which may not become apparent through observation of the individual engaging with enrichment alone. Sakuraba et al. [104] documented the physical rehabilitation and restoration of locomotor function following acute transverse myelitis in a chimpanzee through the application of a computer-based cognitive enrichment task. This task required that the chimpanzee move some distance between the interactive screen and the reward dispenser, effectively utilizing the secondary effect of this enrichment design as a tool for physical rehabilitation. Enrichment may also facilitate the development of social skills [120]. These, along with other secondary effects such as psychological or neurological resilience and reduced reactivity to perturbation, are unlikely to become immediately apparent, as has been suggested by investigations of enrichment in other species [121,122,123]. Such examples reiterate the importance of considering both primary and secondary enrichment effects at a range of temporal scales.

Of the literature considered here, feeding-based approaches to enrichment appear to be the primary method employed (23% of all protocols), followed by enclosure modification (18%). It is not surprising that feeding enrichment is common; feeding is a daily husbandry requirement and thus simple modifications to this procedure in order to generate an enrichment effect would potentially require relatively little effort or financial cost. On the surface, the high prevalence of enclosure modification studies may seem surprising—relative to feeding-based enrichment, modification of an enclosure is not a simple task and is not frequently done. However, given the typically high financial costs of enclosure modifications [15], it is unlikely that enclosure modification would proceed without some efficacy assessment. Enclosure modification has the potential to involve multiple stakeholders, including the institution housing the animals, researchers, funders and/or the general public, and which is likely to generate multiple beneficial outcomes in addition to the benefit to the animals. Such interventions may also provide opportunities for research and education as well as public relations and corporate social responsibility benefits for institutions [124,125] and financial donors alike. In short, despite the high costs involved, enclosure modification has the potential to generate many beneficial outcomes for multiple stakeholders which may explain its prevalence in the literature. Furthermore, the general public tends to perceive naturalistic enclosures as more beneficial for captive animals [126] which would encourage institutions to modify housing accordingly.

The third most common enrichment approaches were social enrichment and the presentation of multiple enrichments simultaneously (combination enrichment). Both of these accounted for a respective 13% of the sample presented here. Most primate species are social to some degree (an estimated 5% are solitary in nature: [127]). Given that one of the aims of enrichment is to promote species-typical behaviour [4], social interventions are a common and important approach to modifying the captive environment, particularly in contexts where opportunities for social engagement may be limited (e.g., laboratories) and reflect that natural social dynamics of most primate species. In some instances, it is possible that caregivers may view social enrichment anthropocentrically whereby they might project human social needs onto captive primates, choosing to apply social enrichment approaches over alternatives, but such an idea remains unexplored in the literature. Regardless of motivation, social enrichment appears to be beneficial (discussed under the Section 4.2) and, when appropriate for the species, reflects the social context of primates in nature, thereby promoting species-typical behaviour and achieving one of the primary aims of environmental enrichment.

The similar prevalence of combination enrichment highlights a scientific problem that prevails in the enrichment literature. As discussed by Swaisgood and Shepherdson [16], despite the potential benefit for the animals, this ‘shotgun approach’ to enrichment is problematic from a scientific perspective because the presentation of multiple enrichment types simultaneously complicates the interpretation of the outcome. Similarly, incomplete or ambiguous qualitative and quantitative reporting (a problem identified in other meta-analyses: [93]), including statistical reporting, or the inappropriate grouping of subjects for analysis (e.g., [76]) complicates interpretation of study findings. We found that such shortcomings undermine the scientific relevance of the findings and make comparisons with other studies difficult. To their credit, many such studies allow for the fine-tuning of existing enrichment protocols by enhancing our understanding of environmental enrichment through novel insights, particularly for under-reported species; consideration of aspects such as individual/species preferences which may be overlooked in other research designs. Nonetheless, future studies have the potential to add more value to their findings and increase their applicability by employing rigorous designs and ensuring that all the available information generated by the study is fully and unambiguously reported.

### 4.2. Enrichment Efficacy in Captive Primates

Generally, our meta-analysis suggest that enrichment has a net beneficial effect for primates, a finding which is in accordance with similar meta-analyses for other animals (mice: [49], pigs: [128], dogs: [129]). The effect appears to be relatively homogenous across the primates (the mean efficacy score for primate enrichment was 0.22 ± 0.02), with the possible exception of the Galagidae, with efficacy index score of 0.00 ± 0.29. The nocturnal galagos [130] exhibit a relatively unique biology and social structure among the primates. Thus, it is possible that the current enrichment practices are not appropriate for the species on biological or ecological grounds. However, the Lorisidae also exhibit a relatively unique biology compared to other primate groups [130] but the publications considered here suggested that enrichment is very effective for this group (efficacy index score of 0.58 ± 0.14). Moreover, given the sample sizes in this meta-analysis (Galagidae: 1 publication, 3 protocols; Lorisidae: 3 publications, 4 protocols) it is difficult to say with any certainty that these interpretations are not merely speculative and, certainly, more research is warranted with these primate groups.

Of the measured biological and ecological factors, only minimum group size in nature emerged as a significant, but weakly associated, predictor of enrichment efficacy. Higher enrichment efficacy scores were calculated for species with a larger minimum group size in nature than those species with smaller minimum group sizes in nature. Neophobia is thought to be mediated through social processes in various species (primates: [131], ungulates: [132]) and thus species that naturally live in larger groups may be able to engage more readily with enrichment through reduced neophobia. However, given the weakness of the association, it may be that for most species the captive environmental context plays a stronger role in determining enrichment efficacy than a species’ biological or ecological characteristics.

Considering the factors of the captive environmental context, the type of enrichment employed emerged as a significant predictor of enrichment efficacy, with training-based enrichment appearing to be the most effective approach. The value of training in animal management is well understood. Training facilitates testing procedures, veterinary care and general day-to-day animal management [133] by allowing animals to voluntarily participate in these procedures and develop associated cognitive skills [134], and thus reduce stress [135]. Human safety has been identified as one of the constraints on enrichment practice [19] and training-based enrichment may mitigate against such risks while providing a benefit to the animal. The mechanism behind the enrichment value of training for primates is unclear (i.e., is the benefit derived from the interaction with a human caregiver, the learning effect of training or the occupation of time on a goal-oriented activity?). Melfi [136] suggests that it is intrinsically enriching for subjects if it stimulates learning and achieves similar effects to other enrichment approaches but that training alone should not form the basis of an enrichment program. Thus, training should be applied in conjunction with other enrichment designs, while avoiding the ‘shotgun approach’ as discussed earlier, in order to maximize the impact on primate subjects.

The presentation of enrichment stimuli which are of little functional relevance to the animals has been highlighted as a shortcoming of current practice [19]. The results of our meta-analysis suggest that this has some support in the literature; object, auditory and olfactory enrichments consistently generated relatively low efficacy scores in this meta-analysis. We suggest that these methods lack dynamicity, are prone to rapid habituation and/or are of limited functional significance for primates in captivity. Enrichment objects are generally static, providing limited stimulation for subjects. For example, several studies in chimpanzees suggest that interest in the objects wanes rapidly but that destructible and manipulable objects are more effective because they afford subjects a degree of control over the object [21,70,137]. Components of destructible objects may offer new interaction opportunities as well (e.g., once torn apart, the pages of a book can be used as nesting material). However, in order to maintain the dynamicity of object enrichment, objects must be changed frequently, presented individually [137] and presented at random, but this has logistical and financial implications [21].

In the case of olfactory and auditory enrichment, the functional relevance of these stimuli for captive primates may underpin their limited efficacy. Olfaction has long been considered as playing a smaller role in primate biology relative to other sensory modalities [138] and evidence suggests that odour cues are of little relevance to captive primates [139]. In addition to complications around the dissipation of odours and interaction of scents [140], habituation to olfactory cues has been reported in captive felids [141], sometimes in a matter of hours [142]. It is conceivable that in environments such as laboratories and zoos where multiple species may be housed in close proximity, individuals habituate to sensory cues which might be of critical importance in nature, such as important resources, potential predators, competitors or conspecifics, but which are of little relevance under captive conditions given their invariable and lingering presence.

Similarly, the relevance of auditory stimuli for captive animals is debatable and studies such as that of Brooker [39] suggest that careful consideration should be given to the biological relevance of such stimuli to captive primates. By contrast, when species-relevant stimuli are applied, enrichment efficacy appears to improve (see [143], for a discussion of sensory-based enrichment and its relevance for captive animals); for example, chimpanzees were more responsive when conspecific food calls accompanied enrichment rather than when human vocalizations were used [144]. Thus, Clark and King [140] argue that in order for enrichment to be effective, there is a need for the presented stimulus to be paired with a meaningful outcome for the animal in order to elicit a response. In support, our findings suggest that it is the potential lack of a real-world consequence of olfactory and auditory enrichment which undermines the efficacy of these approaches to enrichment in primates.

Our findings also appear to concur with the review of sensory-based enrichment of Wells [143]; while no formal analyses were performed, Wells concluded that olfactory and auditory enrichment were of limited efficacy in non-human primates while visual enrichment might be more effective, a view which is mirrored in the efficacy enrichment scores of our study (auditory: 0.02 ± 0.12; olfactory: 0.11 ± 0.05; visual: 0.25 ± 0.16). Moreover, the efficacy of these approaches still appears to be below that of approaches such as training (0.35 ± 0.13), feeding (0.30 ± 0.04) or cognitive enrichment (0.31 ± 0.12). These suggest that, with the possible exception of visual enrichment, sensory approaches to enrichment, while not inherently detrimental, are indeed less likely to be effective for captive primates.

Interaction-based enrichment approaches, considered here as human-initiated direct or indirect inter-specific interactions which are not training-centred or relate to veterinary interventions, appeared to be the only approach that generated a negative efficacy index score, on average (although two of the three studies considered generated positive scores). Thus, such an enrichment approach may be problematic. All three of the reported studies involved interactions between primates and human caregivers. The most obvious objection to such an approach is that it involves unnecessary safety risks, for the human and animal subjects alike, but this approach also assumes that primates experience such interactions as non-threatening. Evidence from zoos suggests that unique dyadic relationships develop between animals and caregivers [145]. However, if interactions involve familiar caregivers who are also involved in invasive experimental or veterinary procedures, subjects may experience their presence as threatening through association. Moreover, primates may perceive many human behaviours as inherently threatening [146] or incomprehensible. Thus, regardless of possible psychological associations subjects may have with specific humans, interactions may be perceived as inherently aversive or uncertain by primates. That three accounts of interaction-based enrichment are present in the peer-reviewed literature suggests that this approach is employed with captive primates and most likely the prevalence in the literature does not reflect the prevalence of the practice. Further examination of this enrichment approach in primates is warranted given the few published studies available. The viability and ethics of interaction-based enrichment in primates should be given careful consideration, especially considering that more effective methods of enrichment which do not necessarily suffer the same pitfalls are available.

Curiously, despite being one of the most prevalent approaches reported in the literature, social enrichment appears to be of a relatively intermediate efficacy (an average efficacy score of 0.21) and comparable to olfactory and ‘other’ enrichment approaches. It is also noteworthy that only 32 protocols testing social enrichment met the criteria for inclusion in our analysis; this may be because the criteria used in this analysis were too conservative in relation to social enrichment studies, social enrichment is widely implemented but seldom the subject of empirical study or that it is not well represented in the peer-reviewed literature, possibly more widely documented in non-peer-reviewed literature (these explanations are not mutually exclusive). The social environment is typically difficult to manipulate in captivity given imposed restrictions from study protocols [147] or risks to both the animals and caregivers associated with social instability in a primate group [50]. Moreover, social interventions may also introduce stress through aggression or dominance-related harassment, which may have ‘spill-over’ effects on other aspects of welfare (e.g., access to food) and is not necessarily appropriate for all species and contexts (reviewed by [45]). However, almost all primates are group-living and even ‘solitary’ species tend to exhibit some degree of sociality (e.g., grey mouse lemurs, *Microcebus murinus* [148]). Thus, maintaining primates under unnatural social conditions, whether alone or in groups, is likely to impact their welfare and study outcomes. Nonetheless, it is notable that social enrichment was not as effective as other commonly employed enrichment strategies (e.g., feeding enrichment).

The social context of enrichment was a significant predictor of enrichment efficacy though. Protocols that did not disclose a social context achieved the highest efficacy scores (0.51), a result which is of limited value other than to further highlight the need for clear and complete reporting. Protocols with socially housed subjects achieved the lowest efficacy scores (0.18), whereas the solitary (0.30) and mixed social contexts (0.22) achieved higher efficacy scores. These scores may be indicative of a social buffering effect whereby individually housed primates derive more benefit from enrichment than socially housed individuals. This is because individuals that are [149] or have been [150] socially housed may be less reactive to stressful stimuli. In other words, individuals which have or previously had exposure to conspecifics may be less stressed. In these cases, enrichment demonstrates marginal beneficial effects and hence a lower apparent efficacy. It is important to note that our analysis considered first-order effects only and it is probable that social context may interact with different enrichment approaches to generate different outcomes. However, given the current limited number of available studies and the pervasive incomplete reporting, this may be better suited to experimental, rather than meta-analytic, examination and the social context of enrichment application must be studied in future.

We found that 68%, 7% and 8% of studies considered concluded a positive, mixed or negligible welfare effect of enrichment on subjects, respectively; none of the studies considered concluded a detrimental effect of an enrichment protocol even though some protocols were associated with negative efficacy index scores (but see case studies identified in Appendix A). On the one hand, this may be the result of publication bias, given that positive publication bias is pervasive and well-documented [151], including specifically in animal welfare [152] and enrichment research [16]. This poses a particular problem for meta-analyses [153]. Alternatively, the interpretations of authors may be intrinsically biased in order to meet a personal desire for improved animal welfare in their study subjects (an individual’s perceptions of animal welfare may be influenced by a range of factors including direct experience and convictions: [154]), a process which can complicate or hinder efforts to improve animal welfare [155]. Also, they may report findings in accordance with the desired outcomes of funders, as suggested by van der Schot and Phillips [152].

It is also possible that the efficacy index used here fails to capture the nuances of the interpretation of the outcomes, which are inherently subjective given the lack of consensus around what constitutes good welfare [113]. Qualitative effects, which might elude statistical detection, may be considered by investigators when drawing conclusions about the effect of an enrichment protocol. This may be particularly important in instances where enrichment results in desirable changes from a management perspective that do not necessarily meet the criteria for improved welfare as described in the literature (e.g., [156]). It is also important to consider that environmental enrichment may prove ineffective for a variety of reasons outside of the enrichment design or apparent environmental context itself. For example, Bielefeldt-Ohmann et al. [157] reported severe clinical pathology only identified following the necropsy of a pig-tailed macaque (*Macaca nemestrina*) which initially appeared unresponsive to environmental enrichment. Satiety may impact engagement with protocols involving food-rewards [158] and the traits of individuals themselves may also impact enrichment efficacy (e.g., social history: [159], age and sex: [160]). Cannon et al. [73] report on attempts to tailor enrichment protocols to the specific needs of individual macaques, yet still found enrichment to be ineffective in some individuals for reasons that were unclear. Thus, the efficacy index used here may fall short of accounting for such effects. However, without a standardized and comparable measure, it is difficult to critically and objectively compare enrichment across contexts and between species.

### 4.3. The Efficacy Index

The efficacy index devised for this meta-analysis was intended as a means of providing a comparative quantification of the impact enrichment protocols have on the animals to which they are applied. It is not intended as a one-size-fits-all metric to be used ‘as is’ but rather as a proof-of-concept tool which offers a standardized metric of welfare impact. Currently, in the absence of such a standardized metric, researchers must make deductions about the effect an enrichment intervention has had on the animals under consideration. This can be challenging: consider, for example, an enrichment intervention which results in reduced stereotypy but increased aggression. All things being equal, which effect should take precedence when drawing conclusions about the overall effect of the enrichment? Certainly, such an assessment will be contextually mediated and thereby the subjective insights of those involved can be invaluable [161]. While most researchers are capable of objectively assessing welfare outcomes, the possibility of bias persists (as discussed in the Section 4.2). Implicit and unconscious biases are well-documented effects in human welfare-related decision making [162,163] and have been implicated in animal welfare decision making too [164]. In light of this, a heavy reliance on subjective or context-dependent interpretation limits applicability in general terms, highlighting the need for a measure which uses widely held standards of welfare in a manner which is comparable. Welfare is a complex concept involving both subjective experiences and objective measures [165] and one which cannot be viewed as a collection of independent factors but rather as a sum of its parts (as illustrated through the Five Domains model of animal welfare: [166]). In alignment with this view, the index provides a generalized cumulative assessment of the welfare significance of enrichment interventions while maintaining the potential to be tailored to specific applications or contexts in order to maximize its relevance and accuracy.

The index is sensitive to the number of reported welfare-relevant measures (*n*). Thus, they may create a skewed impression of effective enrichment for studies that report only those parameters that showed improvement in welfare or report on very few measures of welfare state, resulting in type I errors. Measures of limited relevance to welfare may also create a false interpretation of enrichment efficacy by ‘diluting’ the impact of enrichment, resulting in type II errors. However, as discussed above, for studies that utilize multiple methods of assessing enrichment and which fully report study outcomes, such a tool may provide a relatively simple, objective and comparable estimate of enrichment efficacy.

The ‘ideal’ welfare state in relation to the efficacy index must also be considered. In principle, an efficacy index score of 1 and −1 is indicative of an absolute improvement or deterioration in welfare, respectively, whereby all available welfare measures are indicative of such a change. However, this interpretation hinges on the understanding of what represents improved welfare, an idea for which there is some (but not complete) consensus [113]. The interpretation is likely to be context-specific and may even be unique to individual animals. Measures with a benchmark for comparison (e.g., species-typical C-reactive protein in healthy individuals: [167]) generate a more robust efficacy index than using more subjectively assessed measures (e.g., behavioural diversity). Thus, careful consideration of what constitutes the desired outcome of enrichment is imperative. The index must also be applied and interpreted in context; for example, a training intervention and a foraging device may generate a similar index score despite relying on entirely different mechanisms of action. The variation in modes of action does not invalidate their efficacy but rather highlights the need for careful evaluation and interpretation of the results.

With regard to behavioural measures, it may be useful to consider the activity budgets of free-living conspecific populations as a means of reducing subjectivity. Many studies consider the activity budgets of free-living conspecifics as a good reference against which the activity of captive animals can be assessed in order to estimate welfare (see [168], for a discussion of this approach [169,170,171]). In this paradigm, the proportion of time spent on specific activities by free-living populations could be utilized as a weighting for the measures considered in the index. However, consideration must be given to the fact that some enrichment elicits non-natural (but not abnormal in the context of captivity) behaviour (e.g., using cups for drinking: [172]); these problems can be overcome through careful assessment and application of weighting to welfare measures in the calculation of the index.

We did not assign a weighting to the scores of each of the reported measures because the scope of the study was broad and thus required that assessment be applicable over a wide contextual range. Furthermore, any weighting applied would have been speculative. Thus, the index, as applied here, is conservative in its estimation of welfare impact. This is because it cannot account for the subtle nuances of enrichment effects in specific contexts or for effects that are unreported or incompletely reported. In scenarios where specific metrics may have more impact on welfare than others, applying weighting to the index may provide a more accurate estimate of efficacy. For example, one can argue that self-injurious behaviour is of greater welfare concern than other behaviours, such as inactivity; thus, increasing the weighting of changes in self-injurious behaviour relative to other measures may provide a more appropriate estimation of the overall welfare impact of an enrichment intervention. Furthermore, for studies that report on less generalisable effects, such as quantifying space-use in the context of welfare, enrichments tailored to specific individuals or scenarios in which a specific outcome of enrichment is desirable (if not ‘natural’ sensu stricto or which lies outside the typical criteria for improved welfare), can potentially improve the accuracy of the index when used by applying weighting to context-specific measures. Applied in this manner, the index does potentially provide a useful comparative estimate of enrichment efficacy. The accuracy of the index may be greatly improved through careful consideration of the weighting of specific welfare measures in relation to contextual, individual and species-specific effects in future studies.

## 5. Conclusions

The primary aim of this meta-analysis was to identify approaches to environmental enrichment for captive primates that are most effective. We found that enrichment approaches vary in their efficacy and that approaches that impart a degree of choice and control over the environment for individuals or that are more salient appear to have the highest efficacy. By contrast, sensory-based (with the possible exception of visual enrichment) and object-based approaches appear to offer limited welfare gains for primates. However, these findings should be considered in the context of the current available literature which, despite a trend toward increased numbers of investigations into primate enrichment over time, still only offers a sample of limited size and scope. Further to this, our analysis suggests that there is a positive reporting bias in the literature and thus the efficacy of enrichment as reported here is likely to be similarly biased. There is still much need for further research, particularly for under-represented species and environmental contexts. Future studies should ensure rigorous study design and appropriately detailed reporting in order to maximize the relevance and applicability of their findings. The efficacy index presented here may offer a useful tool to facilitate the comparative critical assessment of enrichment in captive primates and other species. Such an assessment, combined with robust experimental design and accurate reporting, has the potential to improve the practice of environmental enrichment.

## Figures and Tables

**Figure 1 animals-15-00799-f001:**
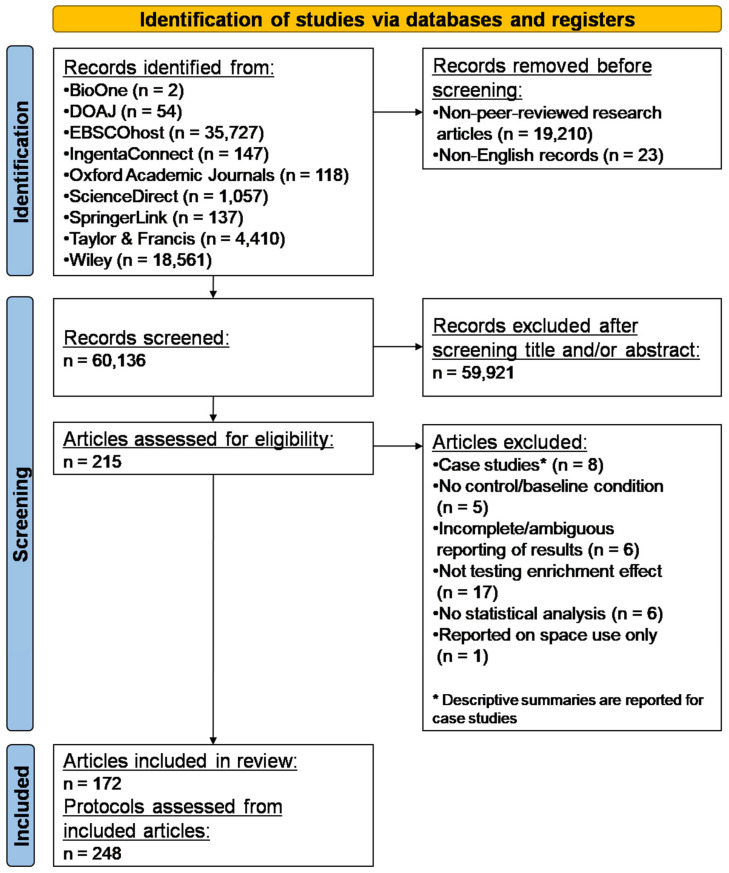
PRISMA 2020 flow chart for selection of studies.

**Figure 2 animals-15-00799-f002:**
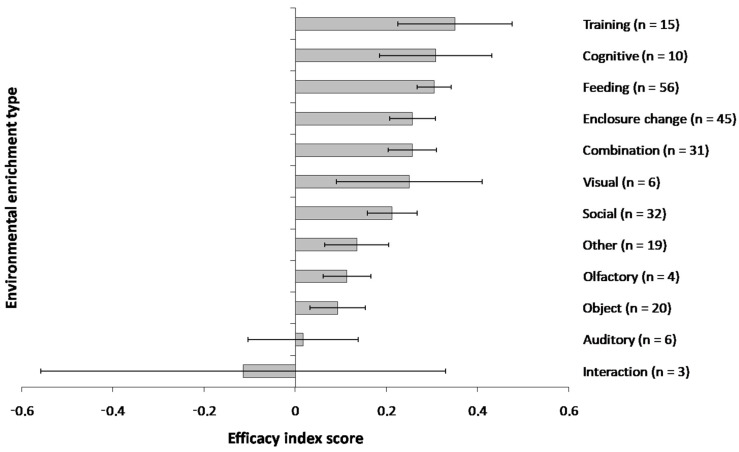
Enrichment efficacy scores for each type of enrichment identified for the meta-analysis based on the available literature up to 31 December 2023. Bars indicate mean efficacy scores and whiskers indicate standard error of the mean. The number of protocols described in the literature (*n*) is presented.

**Figure 3 animals-15-00799-f003:**
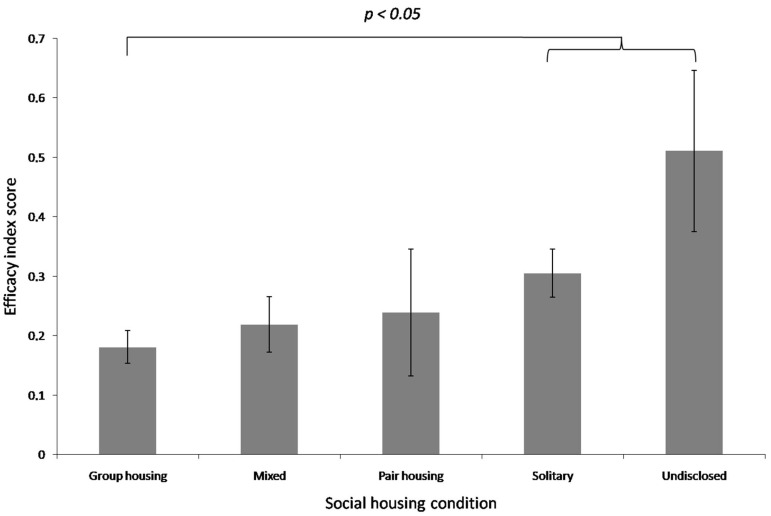
Enrichment efficacy scores for primates housed under various social conditions, namely group housing, mixed conditions (some solitary, some socially housed), pair housing, solitary housing and studies which did not disclose social housing conditions. Bars indicate mean efficacy scores and whiskers indicate standard error of the mean. Statistically significant differences between the enrichment efficacy scores in different housing contexts are indicated by the connecting line above the bars.

**Table 1 animals-15-00799-t001:** Definitions of types of enrichment present in the literature. Definitions are based on functional aspects of the enrichment protocols, including how the enrichment is applied and the intended purpose thereof.

Classification	Definition
Feeding	Protocol designed to alter feeding or foraging behaviour of the subject. Includes presentation of novel foods, foods presented in novel forms or times and foods presented in a manner which requires physical or cognitive effort (excluding interaction with another individual) by subjects in order to gain access.
Object	Protocol involves the presentation of non-nutritive fixed or mobile objects for subjects to interact with which are otherwise absent from the environment and do not pose an overt cognitive challenge.
Olfactory	Protocol presents uncontrollable olfactory stimuli, such as diffused odours or essential oils, as a means of enhancing the sensory environment.
Auditory	Protocol presents uncontrollable auditory stimuli, such as recordings of rainforest sounds or music, as a means of enhancing the sensory environment.
Visual	Protocol presents uncontrollable visual stimuli, such as video or images, as a means of enhancing the sensory environment.
Social	Protocol involves manipulation of social conditions. Includes changes to group size and composition and opportunities for conspecific social interaction.
Cognitive	Protocol presents subjects with an overt cognitive challenge or game which does not involve interaction between conspecific or heterospecific individuals. Food rewards may be involved as incentives to motivate cognition but feeding is not the primary aim.
Training	Protocol involves structured interaction sessions between a human trainer and a primate subject intended to modify the behaviour of the subject, typically to perform a specified behaviour on command.
Interaction-based	Intentional and/or supervised interactions initiated by humans between heterospecific individuals (typically involves gestural or physical interaction between animals and a human) which do not serve an overt training or veterinary function. Interactions may be direct (e.g., physical contact) or indirect (e.g., interaction via a computer-based system or through a protective barrier such as a fence or window).
Enclosure modification	A large-scale qualitative and/or quantitative change in the physical housing environment. Includes the provision of naturalistic elements, novel substrates or structural elements of the housing environment.
Combination	Protocol involves the application of two or more enrichment classifications simultaneously or in a fashion which prevents the effects of the protocol elements from being independently identified.
Other	Any enrichment protocol or design which does not clearly adhere to the classifications above.

**Table 2 animals-15-00799-t002:** Criteria used to assess enrichment protocol efficacy in published literature of environmental enrichment in captive primates. The criteria described reflect widely accepted indicators, as outlined in the contemporary literature, of improved welfare in captive animals.

Domain of Change	Criteria for Beneficial Welfare State
Behavioural	Relative decrease in abnormal behavioural expression (e.g., stereotypy, abnormal posturing or gait, coprophagia).
	Relative increase in ‘desirable’ behavioural expression (e.g., play, social interaction, activity patterns approximating free-ranging conspecifics).
	Relative decrease in pathological behavioural expression (e.g., self-injurious behaviour, trichotillomania).
	Relative decrease in agonistic behavioural expression ^†^ (e.g., aggression, threat display).
	Relative qualitative improvement in behavioural expression (e.g., increased diversity of ‘desired’ behaviour, decreased diversity of abnormal, pathological or aggressive behaviour).
	Relative decrease in self-narcotisation (e.g., reduced self-administration of cocaine or consumption of alcohol).
Neurocognitive	Relatively improved cognitive/perceptual functioning (e.g., problem solving ability, learning and/or memory capacity).
	Relative increase in brain mass and/or neural complexity.
Developmental	Adherence to species-typical physical and psychological/cognitive development or aging.
Physiological/Clinical	Relative decrease in hormonal measures associated with acute and chronic stress as determined through direct measures (e.g., blood sampling) or indirect measures (e.g., urinary or faecal metabolites).
	Relative improvement in physical condition (e.g., increased body mass for the same quantity of food, decreased prevalence of physical pathology or illness). Relative decrease in veterinary interventions and/or time required for veterinary treatment and/or recovery.
	Relative increase in measures of immunological function (e.g., immune cell or cytokine assays) in the absence of infection or disease.
Reproductive	Relative increase in reproductive output.
	Relative decrease in reproductive abnormalities or pathologies (e.g., stillbirths, premature births, congenital defects or disorders).
Survival	Relative reduction in non-age-related deaths, increases in longevity.

^†^ While aggressive behaviour may be considered natural and constitutes an aspect of the behavioural profiles of free-living animal populations, it is generally considered undesirable in captivity from an animal management perspective and for public display scenarios.

**Table 3 animals-15-00799-t003:** Summary table of the distribution of environmental enrichment studies focused on primates in terms of conservation status (IUCN red list) of the species in question.

		Least Concern	Near Threatened	Vulnerable	Endangered	Critically Endangered
Protocols	Total	102	7	21	75	43
	%	41.13	2.82	8.47	30.24	17.34

## Data Availability

The authors will make the data available upon reasonable request.

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
