# Peer review of "A Metric-Based, Meta-Analytic Appraisal of Environmental Enrichment Efficacy in Captive Primates"

_animals, 2025, doi:10.3390/ani15060799_

Round 1
Reviewer 1 Report
Comments and Suggestions for Authors
This manuscript provides a thorough investigation and much needed framework for enrichment assessment and efficacy for captive NHPs. It is well-written, detailed, well-described, and follows logical flow of decision making for the meta-analysis and resulting discussion. The themes addressed are timely, pertinent, and useful in providing guidance for future research on NHP enrichment. I have only a few minor comments. Nice work.
Line 26: suggest a different word instead of “popular.” I don’t think this really conveys what the authors are trying to say. Maybe “common”?
Line 27: suggest a different word instead of “burdened.” This conveys a negative message. Perhaps something simpler, such as “....their captive maintenance includes ethical and husbandry issues...”
Line 78: citation 18 does not seem to provide the most relevant information regarding enrichment here. I would suggest finding a more relevant article (e.g., reference #15, or a more primate-specific publication).
Line 107: suggest rephrasing “does not always work” to “does not always have the intended effect”.
Line 118: please give a brief definition of what is meant by a “predictive approach.”
Line 199 and elsewhere: please check reference
Line 252: please check reference. This seems like it is an important reference given the point it is attempting to illustrate, so I’m curious what is being referenced here.
Line 570: This part of the figure caption is unclear to me. How do the letters denote significant differences? Do they refer to p-values, comparisons across conditions, etc.? Please clarify.
Author Response
This manuscript provides a thorough investigation and much needed framework for enrichment assessment and efficacy for captive NHPs. It is well-written, detailed, well-described, and follows logical flow of decision making for the meta-analysis and resulting discussion. The themes addressed are timely, pertinent, and useful in providing guidance for future research on NHP enrichment. I have only a few minor comments. Nice work.
Authors' response: We thank the reviewer for their complementary appraisal of our manuscript and appreciate the time and effort they have put into reviewing it. We have reviewed the feedback provided and have addressed the reviewer's comments as outlined below.
Line 26: suggest a different word instead of “popular.” I don’t think this really conveys what the authors are trying to say. Maybe “common”?
Authors' response: The reviewer is indeed correct. The word 'common' better conveys our meaning here. We have changed the wording accordingly.
Line 27: suggest a different word instead of “burdened.” This conveys a negative message. Perhaps something simpler, such as “....their captive maintenance includes ethical and husbandry issues...”
Authors' response: We again agree that the term 'burdened' evokes unnecessary and unwanted connotations. We have replaced the word 'burdened' with the word 'involves' instead.
Line 78: citation 18 does not seem to provide the most relevant information regarding enrichment here. I would suggest finding a more relevant article (e.g., reference #15, or a more primate-specific publication).
Authors' response: We have incorporated additional references (including a primate-specific reference as suggested) and altered the wording of the sentence here to better convey our intended meaning. The sentences now read: "Several authors have described diverse categorizations of enrichment approaches [18–20], highlighting the variety of interventions presented in the literature. However, these categorizations may fail to capture the breadth and nuance of enrichment, which ranges from introducing toys and other objects [21,22] through providing artificial grooming partners [23] to playing music to primates [24,25]."
Line 107: suggest rephrasing “does not always work” to “does not always have the intended effect”.
Authors' response: We have modified the text accordingly.
Line 118: please give a brief definition of what is meant by a “predictive approach.”
Authors' response: We understand that we are not necessarily conveying our meaning clearly. We have expanded the point so that the sentence now reads: "However, there is a need for a predictive approach to environmental enrichment research (i.e. an approach which links interventions and outcomes in a consistent manner, to reliably predict how enrichment will affect an animal’s welfare before implementation) in order to increase the relevance and efficacy of enrichment practice [16]."
Line 199 and elsewhere: please check reference
Authors' response: The field code of the reference was corrupted in the conversion of the file to PDF. As such the field code has been removed and replaced with plain text (referring to Table 1) here and throughout.
Line 252: please check reference. This seems like it is an important reference given the point it is attempting to illustrate, so I’m curious what is being referenced here.
Authors' response: As with the previous comment, the field code of the reference was corrupted in the conversion of the file to PDF. As such the field code has been removed and replaced with plain text (referring to Table 2)
Line 570: This part of the figure caption is unclear to me. How do the letters denote significant differences? Do they refer to p-values, comparisons across conditions, etc.? Please clarify.
Authors' response: We recognise that the use of letters to denote the significant differences is not necessarily clear. We have thus modified the graph to use a transverse line above the bars to indicate the significant differences and altered the caption to read: "Statistically significant differences between the enrichment efficacy scores in different housing contexts are indicated by the connecting line above the bars."
Reviewer 2 Report
Comments and Suggestions for Authors
This is an interesting and well-written meta-analysis of studies of environmental enrichment for nonhuman primates maintained in captivity. The authors have identified a number of factors that appear to influence the efficacy of environmental enrichment strategies, including the type of enrichment and a number of external factors, just to name a few. The authors also propose a potentially useful ‘efficacy index’ for enrichment procedures that could be applied when managing primates in captivity. The authors seem to have followed the ‘rules’ for selecting studies for inclusion in a meta-analysis, yet there appear to be numerous studies that could have met their inclusion criteria that were not included. The manuscript is already quite long; therefore it is understandable that not every relevant study could be cited. Overall, there appear to be several paragraphs in the manuscript devoted to tangential issues that could be deleted to shorten the presentation. Additionally, while some of the findings (particularly those related to social enrichment) may be supported by this meta-analysis, they seem to contradict the ‘prevailing wisdom’ of those who actually study and implement environmental enrichment for primates.
Simple Summary
For those working with captive primates, social enrichment is generally considered the most beneficial form of enrichment. It is not clear that this meta-analysis was able to identify social factors as enrichment manipulations. While the ‘rules’ for inclusion of studies in a meta-analysis appear to have been followed, there seem to be a fair number of seemingly appropriate enrichment studies that were not included. There are many citations/references in this fairly long paper, and the inclusion of many additional citations/references will only make the paper even longer. Still, there seems to be some important research that is missing. Additionally, there are numerous passages in the review that discuss studies/protocols that were NOT included in the meta-analysis, which seems like a suboptimal use of page space.
Abstract
Lines 30-31 There have been numerous previous steps toward the development of a predictive enrichment science.
Lines 36-39 While the results of this meta-analysis may suggest that training-based enrichment is the most effective form of enrichment, studies by multiple groups have empirically demonstrated that social enrichment (the opportunity to interact with compatible conspecifics) is typically quite beneficial.
Introduction
Lines 91-106 Many, many more studies of enrichment report benefits of enrichment manipulation(s) than report no effect or detrimental effects. As discussed by the authors later, this could be due to a publication bias against ‘negative’ results, but the clear majority of enrichment studies report enhanced welfare as a function of enrichment.
Lines 107-126 There are at least five different research groups that have made bona fide and useful attempts to address the issues in this paragraph. The works of Bloomsmith, Coleman, Crockett, Novak, Schapiro, and their colleagues are probably worthy of mention here.
Materials and Methods
Lines 166-185 While these selection criteria seem appropriate for inclusion/exclusion of studies in the meta-analysis, it seems as though a fair number of seemingly eligible studies were not included.
Lines 208-216 These sentences seem extraneous.
Lines 225-248 This paragraph is somewhat confusing. While feeding devices may be emptied of food, introduction to a new social partner(s) or new object enrichments will have effects long after the initial introduction.
Lines 263-265 It is unclear why studies in which the outcomes of a metric were not reported or were reported ambiguously were included in the meta-analysis.
Lines 278-281 Is it necessary to state this?
Lines 352-374 It is unclear why two paragraphs are devoted to studies that were NOT included in the meta-analysis.
Lines 393-394 Given the number of statistical tests performed, was there a need to use some sort of correction factor(s)?
Lines 400-402 Would it have made more (statistical) sense, to simply use values of 0, 1, and 2 for the efficacy index, rather than using -1, 0, and 1 and then transforming them by adding one?
Results
Lines 414-429 This number seems fairly low, especially for laboratory studies/protocols.
Lines 435-436 How did “standardized laboratory housing” differ from the other types of housing?
Lines 439-440 Wasn’t one of the criteria for including a study in the meta-analysis some sort of description of a baseline condition?
Figure 2 It is surprising that only 32 protocols used in the analysis involved social enrichment.
Lines 475-483 As the authors suggest, results based on a single enrichment protocol are probably not that meaningful.
Lines 483-500 It is unclear why the authors spend a full paragraph discussing reasons for why studies were not included in the meta-analysis. It might make more sense to focus on aspects of the studies that were included. Were studies that did not completely disclose housing conditions included in the meta-analysis (lines 497-500)?
Lines 507-511 Is this important?
Lines 518-520 As the authors suggest later in the manuscript, this is likely due to a bias against submitting, and attempting to publish, studies with ‘negative results’.
Line 525 A weakly positive relationship is probably not worth mentioning.
Lines 541-552 All of these significant contrasts might be better presented in a simple table.
Lines 553-555 Again, it is unclear how studies with ‘missing data’ (undisclosed social context) could be included in the meta-analysis.
Lines 559-565 The reporting of test statistics and p-values for all of these social context comparisons adds little to the manuscript.
Line 587-601 While the impact of the captive environment on scientific outcomes is likely to be an important consideration for animals maintained in research settings, it is unlikely to be as important a consideration in zoos and other settings. Enrichment’s impact on science practice/validity is an important concern. The authors may be underestimating the broader interpretative perspective in relation to the scientific impact of enrichment.
Lines 623-638 This paragraph is a little bit confusing.
Lines 682-683 It is quite likely that current enrichment studies and programs take baseline severity into account. In many programs, singly housed primates living in cages receive considerably more ‘focused’ enrichment than group-housed animals living in large enclosures. This is one potential reason for why enrichment efficacy as described in this meta-analysis is higher for singly housed primates.
Lines 705-708 While some enrichment STUDIES may be single-faceted in their assessment of enrichment efficacy, many enrichment PROGRAMS take a multifaceted approach to assessing enrichment efficacy.
Lines 709-741 It is unclear why these paragraphs are devoted to case studies, since case studies were not included in the meta-analysis.
Lines 747-761 Many enclosure modification studies are likely to be performed opportunistically. Facilities know that the enclosure will change and are prepared and able to systematically study the effects; they are able to assess efficacy prior to and after the modification.
Lines 762-777 Studies of social enrichment in primates are likely to be considerably more common than this meta-analysis suggests. And as mentioned above, most individuals studying the effects of environmental enrichment, and especially those who are managing captive primates, would agree that social enrichment is the most effective/beneficial/efficacious form of enrichment.
Lines 799-809 Given the small number of publications and protocols for the Galagidae and the Lorisidae, it may not be worth mentioning this information.
Lines 871-873 While the authors’ findings suggest that olfactory and auditory enrichment are of limited efficacy, what data did the authors collect to suggest that this is due to a lack of real-world consequences?
Lines 907-920 It is quite surprising that the meta-analysis suggests that social enrichment is of limited efficacy. There are many, many studies that demonstrate the benefits of social enrichment for primates.
Lines 926-931 This is a fairly obvious statement. Individually housed primates have more ‘empty time’ to fill with enrichment use than do socially housed primates. There are published studies that empirically tested enrichment applications in differing social contexts.
Lines 937 Publication bias certainly could be an issue.
Lines 966-988 The authors may be underestimating the ability of primate enrichment researchers to evaluate and interpret the efficacy of enrichment manipulations and adapt enrichment programs accordingly.
Lines 1031-1035 This is precisely how enrichment managers evaluate the value of enrichment manipulations.
Conclusions
Lines 1058-1060 One would assume that the vast majority of published, peer-reviewed enrichment studies, whether included in this meta-analysis or not, have rigorous study designs and appropriately detailed reporting.
Author Response
This is an interesting and well-written meta-analysis of studies of environmental enrichment for nonhuman primates maintained in captivity. The authors have identified a number of factors that appear to influence the efficacy of environmental enrichment strategies, including the type of enrichment and a number of external factors, just to name a few. The authors also propose a potentially useful ‘efficacy index’ for enrichment procedures that could be applied when managing primates in captivity. The authors seem to have followed the ‘rules’ for selecting studies for inclusion in a meta-analysis, yet there appear to be numerous studies that could have met their inclusion criteria that were not included. The manuscript is already quite long; therefore it is understandable that not every relevant study could be cited. Overall, there appear to be several paragraphs in the manuscript devoted to tangential issues that could be deleted to shorten the presentation. Additionally, while some of the findings (particularly those related to social enrichment) may be supported by this meta-analysis, they seem to contradict the ‘prevailing wisdom’ of those who actually study and implement environmental enrichment for primates.
Authors' response: We thank the reviewer for their complementary appraisal of our manuscript and appreciate the time and effort they have put into reviewing it. We agree that the manuscript is already long and have attempted to shorten the text wherever possible without forgoing the important elements we have presented. Regarding the reviewer’s point around the role of social enrichment, we understand that the finding of our analyses may be surprising but emphasise that we have highlighted that social enrichment is beneficial and should be encouraged as an important tool for animal welfare safeguarding; our findings merely suggest that it may not be as effective as alternative approaches (or potentially as would be expected given the ‘prevailing wisdom’ referenced by the reviewer). We have reviewed the specific feedback provided and have addressed the reviewer's comments as outlined below.
Simple Summary
For those working with captive primates, social enrichment is generally considered the most beneficial form of enrichment. It is not clear that this meta-analysis was able to identify social factors as enrichment manipulations. While the ‘rules’ for inclusion of studies in a meta-analysis appear to have been followed, there seem to be a fair number of seemingly appropriate enrichment studies that were not included. There are many citations/references in this fairly long paper, and the inclusion of many additional citations/references will only make the paper even longer. Still, there seems to be some important research that is missing. Additionally, there are numerous passages in the review that discuss studies/protocols that were NOT included in the meta-analysis, which seems like a suboptimal use of page space.
Authors’ response: [We presume that the feedback provided in relation to the ‘Simple Summary’ section of the manuscript rather relates to the entire manuscript because the reviewer makes reference to citations (which would not ordinarily be included in the ‘Simple Summary’ section as outlined in the ‘Animals’ author guidelines) and missing research and have responded accordingly.] We understand that the reviewer is of the view that social enrichment is the best enrichment approach and we respect this perspective; we would like to emphasise for the reviewer that at no point do we state that social enrichment is not beneficial – on the contrary, we highlight that it is beneficial – but rather that alternative approaches may be more effective and that social context clearly generates nuance to enrichment effects which requires further examination. The reviewer states that the approach taken in this meta-analysis is appropriate. As such, we cannot account for research (legitimate as it may be) that does not meet the criteria for inclusion in the meta-analysis as per the procedures for selection outlined in our manuscript (and which the reviewer appears to agree with). To do otherwise would compromise the integrity of the research by failing to adhere to the PRISMA 2024 guidelines for systematic reviews (a prerequisite for all systematic reviews published in ‘Animals’). Moreover, the reviewer highlights the length of the manuscript as a problem which seemingly precludes the addition of further literature. We have made every effort to address the issues raised in the specific feedback of the reviewer throughout relating to what the reviewer labels as “suboptimal use of page space” and kindly request that the review refer to these responses.
Abstract
Lines 30-31 There have been numerous previous steps toward the development of a predictive enrichment science.
Authors’ response: We acknowledge that ours is not the only effort to move enrichment research into a predictive domain. We have altered the wording here so that it now reads: “As one of the initial steps towards the development of a predictive enrichment science…”
Lines 36-39 While the results of this meta-analysis may suggest that training-based enrichment is the most effective form of enrichment, studies by multiple groups have empirically demonstrated that social enrichment (the opportunity to interact with compatible conspecifics) is typically quite beneficial.
Authors’ response: We do not suggest that social enrichment is not beneficial; our findings suggest that social enrichment is beneficial and we state as much in our discussion (‘4.1. Contexts of application and reporting of enrichment’ , lines 773-777). In the lines referred to by the reviewer, we make reference to the role of social context in influencing enrichment efficacy (i.e. how the social housing conditions of the subjects when enrichment interventions are applied influences how effective the enrichment is in welfare terms), not the merit of social enrichment.
Introduction
Lines 91-106 Many, many more studies of enrichment report benefits of enrichment manipulation(s) than report no effect or detrimental effects. As discussed by the authors later, this could be due to a publication bias against ‘negative’ results, but the clear majority of enrichment studies report enhanced welfare as a function of enrichment.
Authors’ response: We take the reviewer’s point that the majority of reported studies of enrichment describe enhanced welfare outcomes from enrichment interventions. In this section we are highlighting the fact that while enrichment is generally beneficial it is not always beneficial. Given that the majority of (but not all) studies report welfare improvement, it stands to reason that there is room for improvement in both enrichment science and enrichment practice. Further to this, as pointed out by the reviewer, the pervasive publication bias suggests that enrichment interventions which do not generate beneficial outcomes are under-reported, suggesting that the ‘room for improvement’ is likely to be much larger than suggested based on what is reported.
Lines 107-126 There are at least five different research groups that have made bona fide and useful attempts to address the issues in this paragraph. The works of Bloomsmith, Coleman, Crockett, Novak, Schapiro, and their colleagues are probably worthy of mention here.
Authors’ response: We respect the perspective presented here by the reviewer and certainly agree that many researchers have already made significant efforts to improve enrichment practice and science. However, we feel that extending this paragraph with additional referencing exploring the extent of previous research into the field of enrichment science simply to make this point would extend an already lengthy manuscript (a problem that the reviewer has already pointed out) and will add little to the issues that we have already raised in this section. As a compromise we have reworded this section so that it now reads: “Given that, while predominantly beneficial, enrichment does not always have the intended effect, it is important to consider what factors affect enrichment outcomes and how they do so. When considering enrichment practice generally, one major criticism is that it tends to be applied on a trial-and-error basis [19,41]. Additionally, as illustrated through auditory enrichment in Western gorillas, there may be a fallacious perception that if an enrichment protocol has worked in one context, it will work in another. In day-to-day practice, caregivers typically employ those enrichment tools they consider to be effective [32], rather than empirically testing enrichment protocols and the mechanistic dynamics of their application. This is unsurprising, given that organizations rarely possess the personnel and financial resources to apply rigorous investigations of enrichment efficacy [42]; in most institutions, enrichment is approached as an integral component of daily husbandry, rather than empirically tested for efficacy. However, there is a need for a predictive approach to environmental enrichment research (i.e. an approach which links interventions and outcomes in a consistent manner, providing precedence for and foresight into the ways in which enrichment will affect an animal’s welfare) in order to increase the relevance and efficacy of enrichment practice [16]. While many researchers have already made great strides in improving the science of environmental enrichment, further improvement could only make it more structured, objective and reproducible and reduce redundancy, through unnecessary or wasteful utilization of resources that are often already limited. Some studies have attempted to apply predictive approaches to enrichment success [e.g. 43] but far more research is needed. In order to shift enrichment research into a predictive domain, the first step is to identify those enrichment approaches that are effective and why others fail.”
Materials and Methods
Lines 166-185 While these selection criteria seem appropriate for inclusion/exclusion of studies in the meta-analysis, it seems as though a fair number of seemingly eligible studies were not included.
Authors’ response: We have tried to take as conservative approach as possible to the inclusion criteria for the meta-analysis and we feel that, as pointed out by the reviewer here and other reviewers, the selection criteria were appropriate. This was done in order to ensure that the conclusions drawn were supported by the best available evidence, to adhere to the PRISMA 2020 guidelines for systematic reviews and to ensure that our meta-analysis was comparable to similar works in the field of animal welfare. To incorporate other studies on the basis of their merit which would not adhere to the approach we used would compromise the integrity of the systematic review process of PRISMA 2020. We cannot account for studies which were not suggested through the search algorithms, hosted in repositories outside of those we used or which, for whatever reason, did not meet the requirements for inclusion in our sample.
Lines 208-216 These sentences seem extraneous.
Authors’ response: We concur that the inclusion of the description of the slow loris study is not necessary and have therefore removed reference to this work. However, we feel that providing a clear definition for how we identified case studies is necessary because it delineates a criterion which directly affected whether an article was considered for the meta-analysis or not.
Lines 225-248 This paragraph is somewhat confusing. While feeding devices may be emptied of food, introduction to a new social partner(s) or new object enrichments will have effects long after the initial introduction.
Authors’ response: We agree with the point that the reviewer is making here and we feel that this is what is presented in the referenced paragraph. Social partners are indeed more likely to generate lasting impact because their presence provides a consistent and dynamic opportunity to interact (i.e. is enriching), whereas a feeding device is only enriching for as long as there is food available in the device. Hence, we take the stance that accounting for frequency of enrichment presentation may be misleading (in both cases, the frequency would be 1 but the enrichment effect of a feeder is likely to be far less than that of a social partner) and rather consider the duration of direct access (i.e. at any given time, how long was the subject able to engage with the enrichment) when assessing potential predictors of enrichment efficacy.
Lines 263-265 It is unclear why studies in which the outcomes of a metric were not reported or were reported ambiguously were included in the meta-analysis.
Authors’ response: We took this position in order to ensure that all metrics were accounted for when calculating the efficacy index. By excluding those metrics that were ambiguous or unreported, we run the risk of generating bias in the efficacy index (as we discuss later in the section entitled ‘4.3. The efficacy index’, lines 989-997). Unfortunately, many authors selectively report those metrics which generated statistically significant effects rather than presenting the outcomes of all metrics in an equitable manner regardless of statistical valence (a pervasive problem which we discuss in the section entitled ‘3.2. Errant trends in reporting’, lines 484-520) and we wanted to avoid the same problematic approach that we had identified in the existing literature. In light of this, we felt that the most appropriate approach was to assume that no statistically significant change had occurred and applied scoring accordingly.
Lines 278-281 Is it necessary to state this?
Authors’ response: We believe so, yes. It is an important point to clarify, especially for the implementation of the efficacy index. Without this point readers may be left under the mistaken impression that we applied scores for all possible available metrics, not just those specific to each study assessed.
Lines 352-374 It is unclear why two paragraphs are devoted to studies that were NOT included in the meta-analysis.
Authors’ response: We included these paragraphs in the interest of transparency, as per the PRISMA 2020 guidelines for systematic reviews. We have rewritten this section to shorten it and reduced the information about the excluded orang-utan studies to a single line which reads: “Studies which pooled data from different species, but which otherwise met the criteria for inclusion in the meta-analysis, were excluded because these rendered species-level differences in response to enrichment impossible to distinguish.”
Lines 393-394 Given the number of statistical tests performed, was there a need to use some sort of correction factor(s)?
Authors’ response: While there are no strict guidelines around when it is necessary to apply correction factors and there is debate around their universal application, as a general rule corrections should be applied in instances where multiple tests are used to address the same pre-determined hypothesis (i.e. confirmatory studies) but are not necessary for exploratory studies [Althouse. 2016. Adjust for multiple comparisons? It’s not that simple. The Annals of Thoracic Surgery. 101 (5). 1644-1645; Bender & Lange. 2001. Adjusting for multiple testing – when and how? Journal of Clinical Epidemiology. 54 (4). 343-349]. Because we do not raise explicit hypotheses for our meta-analysis, we have adopted an exploratory approach and thus adjustments are not necessary.
Lines 400-402 Would it have made more (statistical) sense, to simply use values of 0, 1, and 2 for the efficacy index, rather than using -1, 0, and 1 and then transforming them by adding one?
Authors’ response: The index was primarily designed using the range of -1 to 1 because this makes the score easily and intuitively interpretable (i.e. a negative number indicates a negative welfare outcome and a positive number indicates a positive welfare outcome). As pointed out by the reviewer and other reviewers, the index provides a practical measure that can be directly applied in a management context; thus providing an index that is intuitive and simple is of the utmost importance in order to maximize its utility in promoting welfare assessment and communication. Statistically speaking, the transformation of the values by adding one has no influence on how they are handled by the generalised linear models used and thus does not affect the statistical outcome.
Results
Lines 414-429 This number seems fairly low, especially for laboratory studies/protocols.
Authors’ response: We would have preferred a larger sample, particularly for those contexts which were not well-represented in the literature. However, the numbers presented are those which comprised the sample following the application of the selection criteria we used.
Lines 435-436 How did “standardized laboratory housing” differ from the other types of housing?
Authors’ response: We acknowledge that the descriptor of ‘standardised’ housing may not be clear to all readers. We have therefore incorporated a footnote in the methods which reads as follows to clarify our meaning: “Standard housing was considered as indoor-only cage rack systems, which typically employ enclosing steel mesh barriers, excretia trays below a steel mesh floor, a feeder hopper and a water supply.”
Lines 439-440 Wasn’t one of the criteria for including a study in the meta-analysis some sort of description of a baseline condition?
Authors’ response: Yes, a baseline condition was a prerequisite for inclusion in the study. We state as such in the section entitled ‘2.1. Literature review and data consolidation’, lines 166-183. In lines 439-440 we are stating that these studies did not provide information about pre-existing enrichment in the baseline condition, not that they did include a baseline condition.
Figure 2 It is surprising that only 32 protocols used in the analysis involved social enrichment.
Authors’ response: We were surprised by how few studies involved social enrichment as well. It may be that social enrichment is utilized widely but is not experimentally evaluated or that it is rather reported widely in the non-peer-reviewed literature and thus is under-represented in the peer-reviewed literature. Alternatively, it may be that such studies are published in journals indexed in databases other than those that were used to produce this manuscript. With that stated, we cannot account for studies which did not appear in our search results or which did not meet the criteria for inclusion. We have incorporated the following to express this point (‘4.2. Enrichment efficacy in captive primates’): “It is also noteworthy that only 32 protocols testing social enrichment met the criteria for inclusion in our analysis; this may be because the criteria used in this analysis were too conservative in relation to social enrichment studies, social enrichment is widely implemented but seldom the subject of empirical study or that it is not well represented in the peer-reviewed literature, possibly more widely documented in non-peer-reviewed literature (these explanations are not mutually exclusive).”
Lines 475-483 As the authors suggest, results based on a single enrichment protocol are probably not that meaningful.
Authors’ response: We agree. That is why we have emphasised that in each case only a single study exists, highlighting the need for further research, and the importance of considering sample sizes and species biases (discussed further in the section entitled ‘4.1. Contexts of application and reporting of enrichment’, lines 602-650).
Lines 483-500 It is unclear why the authors spend a full paragraph discussing reasons for why studies were not included in the meta-analysis. It might make more sense to focus on aspects of the studies that were included. Were studies that did not completely disclose housing conditions included in the meta-analysis (lines 497-500)?
Authors’ response: We have presented this information because these are common practices in reporting, as identified through the literature that we surveyed, which undermine the value and comparability of the studies generally. Incomplete reporting has been highlighted as a problem in other systematic reviews [Lander, N et al. 2017. Characteristics of teacher training in school-based physical education interventions to improve fundamental movement skills and/or physical activity: a systematic review. Sports Medicine. 47(1). 135–161] and we are raising the issue here because it has the potential to undermine the applicability and validity of the science being presented by introducing the possibility of bias in the publications or omission of important outcomes. We illustrate the problem with incomplete reporting when discussing the role of social context in predicting enrichment efficacy (the section entitled ‘4.2. Enrichment efficacy in captive primates’, lines 921-932). With regard to reporting of housing conditions, none of the included studies failed to mention any housing conditions but many studies included in the analysis failed to disclose at least one aspect of housing conditions (e.g. social conditions, housing type or housing state).
Lines 507-511 Is this important?
Authors’ response: We have highlighted this because reporting of precise p-values promotes transparency and conforms to widely held standards of statistical reporting (e.g. American Psychological Association. 2024. APA Style numbers and statistics guide. https://apastyle.apa.org/instructional-aids/numbers-statistics-guide.pdf). It is particularly important when considering results which are statistically not significant; categorically reporting such results as ‘NS’ or ‘p > 0.05’ potentially masks trends toward significance. For example, a factor which generates a p-value of 0.053 would be considered as not statistically significant but could indicate a biologically important and relevant effect. In the context of animal welfare, this might result in the dismissal of an effect which may have a meaningful impact on the animals themselves or prevent readers from identifying key factors for further research and exploration. To explain our rationale, we have incorporated the following: “such practice fails to conform to widely held standards of statistical reporting [84] and is particularly problematic for statistically non-significant results because categorical reporting may mask biologically meaningful effects or trends which are not strictly statistically significant”
Lines 518-520 As the authors suggest later in the manuscript, this is likely due to a bias against submitting, and attempting to publish, studies with ‘negative results’.
Authors’ response: We agree with the reviewer. Publication bias is a likely explanation for why few studies reported negative outcomes of enrichment interventions. A section of the discussion is dedicated to exploring this specifically (under the section entitled ‘4.2. Enrichment efficacy in captive primates‘, lines 933-946).
Line 525 A weakly positive relationship is probably not worth mentioning.
Authors’ response: The generalised linear modeling identified minimum group size as a statistically significant predictor of efficacy and we feel that fully interrogating the data is necessary to ensure meaningful interpretation of the outcomes, hence the correlation was performed and reported on. We report it here because incomplete statistical reporting is a pervasive problem that we identified in the existing literature (as discussed in the section entitled ‘3.2. Errant trends in reporting’, lines 501-520) and we feel that we do not wish to perpetuate this reporting practice. We go on to discuss this in the discussion section (‘4.2. Enrichment efficacy in captive primates’, 810-819).
Lines 541-552 All of these significant contrasts might be better presented in a simple table.
Authors’ response: The contrasts are presented in Table S5. As such, and in an effort to reduce the length of the text, we have removed the statistical reporting here and directed the reader to Table S5 for all contrasts rather than only the non-significant effects.
Lines 553-555 Again, it is unclear how studies with ‘missing data’ (undisclosed social context) could be included in the meta-analysis.
Authors’ response: As we outline in the section entitled ‘2.1. Literature review and data consolidation’ (lines 205-207), not all studies provided the full complement of data that we extracted from the literature. For example, some might disclose the housing type and pre-existing enrichment but fail to disclose whether subjects were housed solitarily, pair-housed or group-housed. Thus, we incorporated the ‘Undisclosed’ category to account for these missing data. The prevalence of this practice is also part of why we included the section entitled ‘3.2. Errant trends in reporting’ (lines 484-520).
Lines 559-565 The reporting of test statistics and p-values for all of these social context comparisons adds little to the manuscript.
Authors’ response: Incomplete statistical reporting is a common problem that we identified in the literature (as discussed in the section entitled ‘3.2. Errant trends in reporting’, lines 501-520) and we wanted to avoid the same problematic approach. We thus have chosen to report our statistical results in full in the spirit of transparency and openness.
Line 587-601 While the impact of the captive environment on scientific outcomes is likely to be an important consideration for animals maintained in research settings, it is unlikely to be as important a consideration in zoos and other settings. Enrichment’s impact on science practice/validity is an important concern. The authors may be underestimating the broader interpretative perspective in relation to the scientific impact of enrichment.
Authors’ response: We take the point of the reviewer here that the context in which enrichment is applied is likely to determine the relative importance of the scientific impact. To this end we have incorporated the following line to illustrate this point: “While the scientific implications of enrichment interventions are likely to be more consequential in a research context than a husbandry context, a strong argument can be made for the need to ensure scientific validity in captive animal research generally [4].”
Lines 623-638 This paragraph is a little bit confusing.
Authors’ response: We have reworded the text to clarify our meaning. It now reads: “Some species may be viewed as less charismatically appealing, of lesser conservation significance (thereby seemingly warranting less attention) or are less likely to generate funding for primate enrichment research. Fundraising-centred investigations suggest that both charismatic appeal and similarity to humans significantly impact on the perceived value or degree of interest in a species [98] and, to this end, species or families that are arguably more human-like (e.g. great apes) attract more interest [21]. This results in an overstated prevalence of these groups in the literature, as noted in a previous meta-analysis on enrichment in zoos [16]. Furthermore, the general public exhibits distinct species-viewing preferences [96,99] and influence how enrichment is practiced in zoos [100]. We found 60 published protocols that focused on four species within the Hominidae (an average of 15 protocols per species) whereas 107 published protocols focused on 22 species within the Cercopithecidae (an average of 4.9 publications per species). This indicates a proportional bias towards species within Hominidae [16,also previously identified as a particularly popular group in enrichment research: ,101]. Assuming that the distribution of species in the literature mirrors their prevalence in captivity, this potentially implicates charismatic appeal and ‘human-ness’ in determining whether a species is the focus of enrichment research.”
Lines 682-683 It is quite likely that current enrichment studies and programs take baseline severity into account. In many programs, singly housed primates living in cages receive considerably more ‘focused’ enrichment than group-housed animals living in large enclosures. This is one potential reason for why enrichment efficacy as described in this meta-analysis is higher for singly housed primates.
Authors’ response: We agree that studies may be taking baseline severity into account. However, this does not mean that future studies should not account for baseline severity. The latter is the point we are making here.
Lines 705-708 While some enrichment STUDIES may be single-faceted in their assessment of enrichment efficacy, many enrichment PROGRAMS take a multifaceted approach to assessing enrichment efficacy.
Authors’ response: Our focus was on studies reported in the literature and thus we are not in a position to comment on the role of enrichment programmes. This would lie outside the scope of our analysis.
Lines 709-741 It is unclear why these paragraphs are devoted to case studies, since case studies were not included in the meta-analysis.
Authors’ response: We have included this section to highlight the valuable contribution that case reports can make to enrichment practice. While case reports did not meet the criteria for inclusion in the analysis, the case reports discussed illustrate some of the risks associated with enrichment in primates that would otherwise have gone unreported. In the context of a field where publication bias is evident, highlighting these findings is that much more important. In an effort to present a transparent and unbiased view of current enrichment practice we feel that it is critical that such outcomes be highlighted in order to meaningfully appraise primate environmental enrichment.
Lines 747-761 Many enclosure modification studies are likely to be performed opportunistically. Facilities know that the enclosure will change and are prepared and able to systematically study the effects; they are able to assess efficacy prior to and after the modification.
Authors’ response: We agree with the reviewer here. We are making the point that an assessment of the enclosure change is highly likely because there are considerable costs (financial and manpower) associated with the restructuring of enclosures and that such assessments stand to benefit multiple stakeholders.
Lines 762-777 Studies of social enrichment in primates are likely to be considerably more common than this meta-analysis suggests. And as mentioned above, most individuals studying the effects of environmental enrichment, and especially those who are managing captive primates, would agree that social enrichment is the most effective/beneficial/efficacious form of enrichment.
Authors’ response: While we respect the reviewer’s viewpoint that social enrichment is the most effective enrichment approach, our findings do not support this view. Our findings suggest that social enrichment is certainly an effective enrichment approach but that alternative approaches (namely training, cognitive, feeding, enclosure change, combination and visual enrichment approaches) are equally or even more effective. We understand that the prevailing perspective amongst many researchers and those in the animal husbandry field may be that social enrichment is the best approach but we can only report on what our analyses showed. Please also bear in mind that our analysis focused on first-order effects only and so it is entirely possible that interactions between social context and type of enrichment have a major influence on enrichment efficacy (we devote much discussion to the potential role of social context in the section entitled: ‘4.2. Enrichment efficacy in captive primates’, lines 907-932; to this end we have also added the following to the end of the paragraph: “It is important to note that our analysis considered first-order effects only and it is probable that social context may interact with different enrichment approaches to generate different outcomes. However, given the current limited number of available studies and the pervasive incomplete reporting, this may be better suited to experimental, rather than meta-analytic, examination and the social context of enrichment application must be studied in future.”).
Lines 799-809 Given the small number of publications and protocols for the Galagidae and the Lorisidae, it may not be worth mentioning this information.
Authors’ response: We are specifically mentioning these groups to highlight that there is the possibility that ecological and evolutionary factors may be influencing the effects of enrichment in a captive context but that the lack of research into these specific groups makes it difficult to determine whether this is indeed the case. This is important in highlighting the potential role of ecology and evolution in affecting captive animal maintenance, factors which have been highlighted as important for other species [e.g. Clubb & Mason. 2007. Natural behavioural biology as a risk factor in carnivore welfare: How analysing species differences could help zoos improve enclosures. Applied Animal Behaviour Science. 102 (3-4). 303-328.] and which underpin the principle of biological relevance in enrichment application, and emphasises the need for further research into enrichment practice and application.
Lines 871-873 While the authors’ findings suggest that olfactory and auditory enrichment are of limited efficacy, what data did the authors collect to suggest that this is due to a lack of real-world consequences?
Authors’ response: While we did not actively collect data to test whether real-world relevance may underlie the relatively poorer efficacy of olfactory and auditory stimuli for primate enrichment (which would be beyond the scope of our study), this view is discussed in the literature (as we point out in lines 862-871 of the same paragraph referred to by the reviewer). We are merely echoing a perspective which has already been expressed by other authors and which is supported by our empirical analysis of the studies reported in the literature.
Lines 907-920 It is quite surprising that the meta-analysis suggests that social enrichment is of limited efficacy. There are many, many studies that demonstrate the benefits of social enrichment for primates.
Authors’ response: We acknowledge that the phrasing of this statement may have created the wrong impression. We have changed the wording so that it now reads: “…social enrichment appears to be of a relatively intermediate efficacy…”
Lines 926-931 This is a fairly obvious statement. Individually housed primates have more ‘empty time’ to fill with enrichment use than do socially housed primates. There are published studies that empirically tested enrichment applications in differing social contexts.
Authors’ response: The point that the reviewer makes that the view expressed is supported by published studies is correct in our opinion and is supported by our findings.
Lines 937 Publication bias certainly could be an issue.
Authors’ response: We agree with the reviewer. In line with this view, we state that publication bias is a likely explanation for the effect (lines 937-939).
Lines 966-988 The authors may be underestimating the ability of primate enrichment researchers to evaluate and interpret the efficacy of enrichment manipulations and adapt enrichment programs accordingly.
Authors’ response: As stated in lines 977-982 of the paragraph in question, relying purely on the subjective assessment of enrichment outcomes by individuals increases the likelihood of bias and potentially limits applicability of findings. We do not suggest that researchers are poor at objectively assessing enrichment outcomes. To clarify our position, we have reworded the paragraph so that it now reads: “The efficacy index devised for this meta-analysis was intended as a means of pro-viding a comparative quantification of the impact enrichment protocols have on the animals to which they are applied. It is not intended as a one-size-fits-all metric to be used ‘as is’ but rather as a proof-of-concept tool which offers a standardised metric of welfare impact. Currently, in the absence of such a standardised metric, researchers must make deductions about the effect an enrichment intervention has had on the animals under consideration. This can be challenging: consider, for example, an enrichment intervention which results in reduced stereotypy but increased aggression. All things being equal, which effect should take precedence when drawing conclusions about the overall effect of the enrichment? Certainly, such an assessment will be contextually mediated and thereby the subjective insights of those involved can be invaluable [160]. While most researchers are capable of objectively assessing welfare outcomes, the possibility of bias persists (as discussed in the ‘Enrichment efficacy in captive primates’ sub-section). Implicit and unconscious biases are well-documented effects in human welfare-related decision making [161,162] and have been implicated in animal welfare decision-making too [163]. In light of this, a heavy reliance on subjective or context-dependent interpretation limits applicability in general terms, highlighting the need for a measure which uses widely held standards of welfare in a manner which is comparable. Welfare is a complex concept involving both subjective experiences and objective measures [164] and one which cannot be viewed as a collection of independent factors but rather as a sum of its parts [as illustrated through the Five Domains model of animal welfare: ,165]. In alignment with this view, the index provides a generalised cumulative assessment of the welfare significance of enrichment interventions while maintaining the potential to be tailored to specific applications or contexts in order to maximise its relevance and accuracy.”
Lines 1031-1035 This is precisely how enrichment managers evaluate the value of enrichment manipulations.
Authors’ response: We agree and we are illustrating that such rationale can be formalized and accounted for by applying weighting factors to the index.
Conclusions
Lines 1058-1060 One would assume that the vast majority of published, peer-reviewed enrichment studies, whether included in this meta-analysis or not, have rigorous study designs and appropriately detailed reporting.
Authors’ response: While we respect the reviewer’s perspective that most published, peer-reviewed enrichment studies involve rigorous study design and appropriately detailed reporting and certainly would hope this to be the case, the findings of our analysis do not fully support this view. While study designs were not assessed in this analysis and are therefore outside of the scope of this review, the reporting in many published, peer-reviewed studies was frequently found to be lacking in sufficient detail to allow general comparison as evidenced by the need for the inclusion of the ‘Undisclosed’ category in our analyses. On this basis alone, encouraging future studies to adopt sound, declarative reporting practices, including complete and transparent reporting, can only improve the field and result in more meaningful outcomes for future research and the animals involved.